# Live-cell p53 single-molecule binding is modulated by C-terminal acetylation and correlates with transcriptional activity

Alessia Loffreda[1,2], Emanuela Jacchetti[1,7], Sofia Antunes[1], Paolo Rainone[1,3], Tiziana Daniele[1], Tatsuya Morisaki [4,8], Marco E. Bianchi[5,6], Carlo Tacchetti[1,6] & Davide Mazza[1,2]

Live-cell microscopy has highlighted that transcription factors bind transiently to chromatin but it is not clear if the duration of these binding interactions can be modulated in response to an activation stimulus, and if such modulation can be controlled by post-translational modifications of the transcription factor. We address this question for the tumor suppressor p53 by combining live-cell single-molecule microscopy and single cell in situ measurements of transcription and we show that p53-binding kinetics are modulated following genotoxic stress. The modulation of p53 residence times on chromatin requires C-terminal acetylation —a classical mark for transcriptionally active p53—and correlates with the induction of transcription of target genes such as *CDKN1a*. We propose a model in which the modification state of the transcription factor determines the coupling between transcription factor abundance and transcriptional activity by tuning the transcription factor residence time on target sites.

[1] Istituto Scientifico Ospedale San Raffaele, Centro di Imaging Sperimentale, Milano 20132, Italy. [2] Fondazione CEN, European Center for Nanomedicine, Milano 20133, Italy. [3] Institute of Molecular Bioimaging and Physiology, National Researches Council, Segrate 20090 (MI), Italy. [4] Fluorescence Imaging Group, National Cancer Institute, NIH, Bethesda, Maryland 20892, USA. [5] Istituto Scientifico Ospedale San Raffaele, Chromatin Dynamics Unit, Milano 20132, Italy. [6] Università Vita-Salute San Raffaele, Milano 20132, Italy. [7] Present address: Dipartimento di Chimica, Materiali e Ingegneria Chimica "G.Natta". Politecnico di Milano, Piazza Leonardo da Vinci 32, Milano 20133, Italy. [8] Present address: Department of Biochemistry and Molecular Biology, Colorado State University, Fort Collins, CO 80523, USA. Alessia Loffreda and Emanuela Jacchetti contributed equally to this work. Carlo Tacchetti and Davide Mazza jointly supervised this work. Correspondence and requests for materials should be addressed to C.T. (email: tacchetti.carlo@hsr.it) or to D.M. (email: mazza.davide@hsr.it)

I nducible transcription in response to an activating stimulus is mediated by the activation of transcription factors (TFs) and by their binding to responsive elements (REs) of enhancers and/or promoters of the target genes. TF activation is often achieved through two main mechanisms: (i) the increase of the TF levels in the nucleus and (ii) the induction of post-translational modifications (PTMs) that render the TF transcriptionally competent[1–3]. The role of an increase in nuclear TF levels on transcriptional activation has been widely discussed as it would favor a more frequent association between the TF and the REs, presumably resulting in enhanced transcriptional activation[4–6]. On the other side, the roles of TF PTMs are more difficult to predict as they might impact multiple processes ranging from TF/DNA affinity, TF/cofactor interactions to TF degradation rates[2].

Mounting evidence indicates that TF/DNA association are generally transient and a few studies have begun to suggest that the duration of the TF/REs interaction (the residence time of the TF) might be an important parameter to regulate transcription[3, 7, 8]. However, it remains unclear whether these recent studies focusing on TF kinetics can be connected to classical studies that identified the role of PTMs of the TF and of co-factors on transcriptional activation.

In this paper we investigate the connection between transient residence times, PTMs and transactivating capability of TFs, by focusing on the tumor suppressor p53, a TF that plays a central role in controlling the cellular response to distinct stress signals, and engaging transcriptional programs leading to cell-cycle arrest, senescence or apoptosis, depending on the nature and on the strength of the offending stimulus[9, 10]. In response to genotoxic stress, p53 expression levels increase[11], due to the inhibition of the interaction of p53 with its negative regulator MDM2, which directs p53 to degradation. This increase in p53 levels is not sufficient to induce transcription of target genes, indicating that some other properties of the TF, other than its abundance must encode messages responsible for transcriptional activation[12]. Biochemically, the modulation of p53-mediated transcription has been associated to PTMs such as C-terminal domain (CTD) acetylation, but it is not clear how these modifications are translated into a different physical behavior of the p53 protein: one possibility is that CTD acetylation modulates the p53 affinity for its REs on DNA. While in vitro results seem to support this hypothesis[13], results obtained in the cellular milieu by chromatin immunoprecipitation (ChIP) are more controversial: CTD-acetylated p53 accumulates at active transcription sites in response to stress signals, but so do mutants with impaired CTD acetylation[14]. Further, recent genome-wide ChIP studies have shown that mutants mimicking p53 acetylation are not recruited to REs more efficiently than unacetylated wild-type p53[15].

Also, according to immunoprecipitation experiments, p53 binding to REs seems mainly tuned by modulating p53 expression levels, rather than by PTMs, as in response to DNA damage the fold-change in p53 occupancies closely match the changes in the TF expression[16].

A possible explanation of these discrepancies arises from the recent observation that the typical occupancy measurements obtained by conventional ChIP on TFs are a poor predictor of transcriptional activity, while the TF residence time correlates better with transcriptional activation[7], in particular for factors that undergo rapid association and dissociation[17–19]. Indeed, single molecule and ensemble-average live-cell fluorescence microscopy has revealed that the interactions of p53 with DNA are transient (on the time-scale of seconds) both at specific and at non-specific targets[20–23]. Unfortunately, all live-cell measurements of p53 binding have been obtained in

unstimulated cells only, leaving unanswered whether the binding kinetics of p53 are modulated upon the infliction of DNA damage.

Here, we combine live-cell single-molecule tracking, population-based and single-cell-based measurements of transcription to provide evidence that the p53 residence time on REs is a physical read-out of the acetylation state of p53-CTD, which correlates with p53-mediated transcriptional activity. To this end we show that (i) the p53 residence time following activation by genotoxic stress is lengthened; (ii) the p53 residence time is controlled by the acetylation state of the p53-CTD, and that (iii) the p53 live-cell residence time is a better predictor of its transcriptional activity than p53 abundance. Our results point to a model in which the acetylation of p53 would act as a clutch that can switch on and off the coupling between p53 levels and transcriptional activity in single cells: in basal conditions (low-binding affinity) an increase in p53 concentration does not result in the transactivation of target genes such as the cell-cycle arrest CDKN1a gene, but a modulation in p53 acetylation is also needed, impinging on its binding kinetics. More generally, our results provide a direct connection between chemical modifications of the TF and its kinetic behavior, and lead us to propose a simple "modification-driven kinetics" model in which PTMs control the transactivating capability of the TF by tuning its residence time on REs.

## Results

**p53 bound fraction increases following DNA damage.** In order to study the interplay between p53 expression, p53-binding kinetics and transcriptional activation in single cells, we generated the stable cell line MCF-7/6/Hp53 expressing HaloTag-p53, under the control of a Tet-regulated promoter (see Methods). Post-translational labeling systems such as HaloTag are well-suited for single-molecule imaging approaches as they allow to label an arbitrarily small subpopulation with bright and photostable organic dyes[24]. Unless otherwise stated, we exploited the leakiness of the Tet-regulated promoter, to minimize the overexpression of tagged p53. We verified that HaloTag-p53 could activate the transcription of known p53 target genes upon long-term treatment with doxycycline (Supplementary Fig. 1) and that DNA damage obtained with ionizing radiation (IR, 10 Gy gamma rays) could induce HaloTag-p53 stabilization (Fig. 1a). Similar results were found when HaloTag-p53 was inducibly expressed over a p53 null background (Supplementary Fig. 1). Fluorescence microscopy revealed that IR resulted in an increase of p53 expression and localization in the nucleus (Fig. 1b) 2 h after the genotoxic insult, although we noticed that even in basal conditions a fraction of cells (< 10%) displayed high HaloTag-p53 nuclear levels, similar to what we observed by immunostaining of the endogenous p53 in the parental cell line (Fig. 1b). Previous studies[12] revealed that these cells are those responding to physiological sources of stress and have an attenuated transcriptional response compared to cells responding to exogenous stress as IR: we therefore aimed at measuring the differences between the binding kinetics of tagged p53 in cells expressing high TF levels in basal conditions and cells exposed to ionizing radiation.

A modulation of affinity of p53 for its cognate sites might arise by an increase in the efficiency of the TF search mechanism (resulting in a shorter search time for cognate sites) or by an increase in the stability of the interactions between the TF and the cognate site (resulting in longer residence times at the cognate site). As in both cases the net result would be an increase in the fraction of chromatin-bound p53 molecules, we first tested whether the p53 bound fraction was modulated following

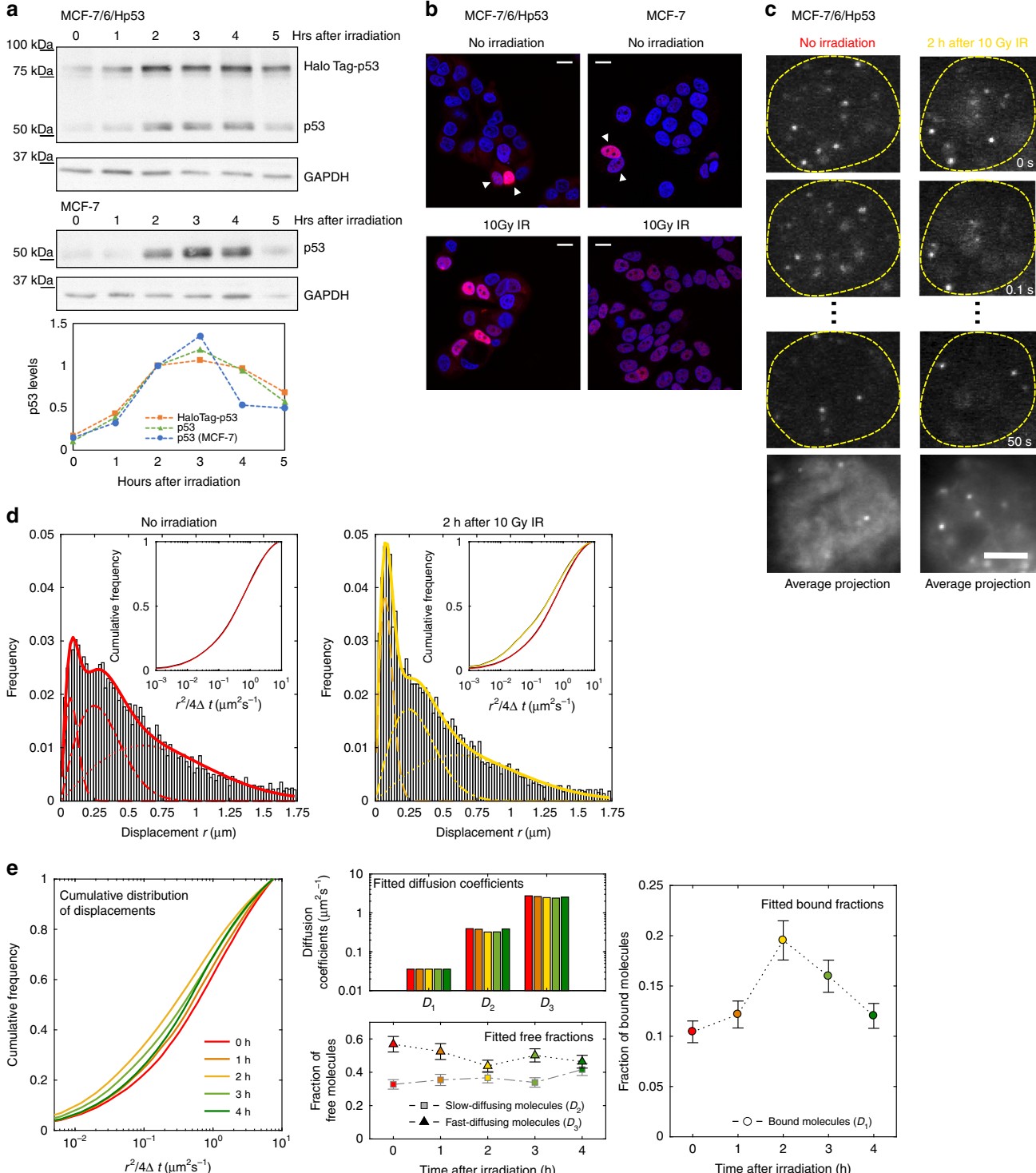

**Fig. 1** Single-molecule tracking of HaloTag-p53 in response to DNA damage. **a**, **b** Characterization of MCF-7/6/Hp53 cell line **a** Western blot of p53 and HaloTag-p53 at different times following the induction of DNA damage by 10 Gy IR in MCF-7/6/Hp53, a stable cell line expressing HaloTag-p53, and in the parental breast cancer cell line MCF-7 (2 replicates). **b** Representative confocal microscopy fields of MCF-7/6/HaloTag-p53 cells before and after exposure to 10 Gy IR. HaloTag-p53 was labeled with HaloTag-TMR fluorescent ligand. A small fraction of cells displays high HaloTag-p53 levels even when unstimulated. *Scale bar* 15 μm. A similar fraction of p53-positive cells can be identified in the parental cell line by immunofluorescence. *Scale bar* 15 μm. **c**–**e** Single-molecule tracking of HaloTag-p53. **c** The average projection of the images allows the identification of the cell nucleus and of the sites of relatively stable immobilization of the single molecules, which appear as bright spots. *Scale bar* 5 μm. **d** The movies were tracked to compute the distribution of single-molecule displacements ($n_{cells}$ = 8, $n_{displacements}$ = 8876 for 0 h and 7973 for 2 h). The distributions were fitted with a three-component diffusion model, where the slowest diffusion component is representative of chromatin-bound molecules. The *insets* show the cumulative distribution of displacements. Fitted parameters are shown in Table 1. **e** The experiments above were repeated at different times following the induction of DNA damage to measure the diffusion coefficients and the fraction of molecules in the bound state and in the two free states (3 replicates, $n_{cells}$ = 17, 9, 23, 26, 22, $n_{displacements}$ = 18 609, 8704, 41 854, 38 155, 22 889 for 0, 1, 2, 3, and 4 h after DNA damage, *error bars*: SD)

**Table 1 Parameters obtained from fits of the distribution of displacements for HaloTag-p53 shown in Fig. 1d**

| Parameters | No irradiation | 2 h after 10 Gy IR |
|---|---|---|
| $D_1$ ($\mu m^2 s^{-1}$) | $0.014 \pm 0.008$ | $0.014 \pm 0.008$ |
| $D_2$ ($\mu m^2 s^{-1}$) | $0.29 \pm 0.02$ | $0.29 \pm 0.02$ |
| $D_3$ ($\mu m^2 s^{-1}$) | $1.85 \pm 0.07$ | $1.78 \pm 0.07$ |
| $f_1$ (bound fraction) | $0.102 \pm 0.013$ | $0.21 \pm 0.02$ |
| $f_2$ | $0.35 \pm 0.04$ | $0.34 \pm 0.04$ |
| $f_3$ | $0.55 \pm 0.06$ | $0.45 \pm 0.05$ |

activation by DNA damage. To this end, we imaged p53 at the single-molecule level in cells displaying detectable nuclear levels of the protein, before and after irradiation. MCF-7/6/Hp53 cells were incubated with sub-nanomolar concentrations of HaloTag-TMR and washed extensively to remove the unbound ligand. We excluded the possibility that unconjugated ligand could bias our analysis by performing the labeling protocol on parental cells that do not express HaloTag-p53 (Supplementary Fig. 2). We observed the fluorescently tagged molecules with a microscope equipped with highly inclined optical sheet illumination[25], as we previously described[22]. To follow individual molecules for prolonged times without excessive photobleaching, we adopted stroboscopic illumination: we collected time-lapse movies at a rate of 10–25 frames per second (fps) with a laser exposure of 5 ms for each image (Fig. 1c). Single molecules were tracked with our previously described tracking software[22] as described in the Supplementary Methods.

The single-molecule movies (Supplementary Movies 1 and 2) featured a larger fraction of immobilized p53 molecules following DNA damage, as quantified by computing the distribution of single-molecule displacements between consecutive frames (Fig. 1d). These immobilized, chromatin-bound molecules contribute to the distribution of displacements with a slow diffusion coefficient component ($< 0.1 \, \mu m^2 s^{-1}$), caused by the limited precision in the localization of individual molecules and by the movement of chromatin itself. The relative amplitude of this slow component can be used to quantify the fraction of bound molecules[22, 26]. By fitting the distribution of single-molecule displacements with a three-component diffusion model, we estimated the diffusion coefficient for the slowest component $D_1 = 0.014 \, \mu m^2 s^{-1}$ (Table 1), corresponding to an average displacement of $r = \sqrt{4D_1 t} \sim 75\,nm$, in good agreement with the distance covered by chromatin-bound proteins in 0.1 s[22, 27]. From this analysis, we determined that the fraction of p53 bound to chromatin increased from 10% in undamaged cells to 21% 2 h after irradiation (Fig. 1d and Table 1).

We point out that the relatively slow acquisition rate used in this work might artifactually impact the correct measurement of the diffusion properties of free molecules.

To exclude the possibility that slow acquisition rates could also cause errors in the estimation of the fraction of chromatin-bound p53, we performed experiments using a 10× faster acquisition (100 fps, exposure 2 ms) and measured comparable bound fractions (Supplementary Fig. 2c). A qualitatively similar binding modulation was also found when HaloTag-p53 was stably expressed over a p53 null background (Supplementary Fig. 3). By repeating the analysis of single-molecule displacements at different time points following the induction of DNA damage, we found that the diffusion coefficients of p53 remained unchanged along the time-course but the fraction of molecules in the bound state transiently increased, peaking at 2 h post IR and decreased again at later times (Fig. 1e).

**Specific p53 residence time increases following activation.** The modulation of the p53 chromatin-bound fraction can occur in two non-exclusive ways: following DNA damage p53 might increase its binding time to cognate sites and/or it might decrease its free time between binding events. To distinguish between these two possibilities, we measured the duration of binding events by single-molecule imaging. To provide an unsupervised assessment of p53 residence times, we analyzed kymographs of the single-molecule movies: in a kymograph, an immobilized p53 molecule would be seen as a straight segment parallel to the temporal axis, and the distribution of residence times can be quantified by measuring the length of these segments (Fig. 2a). We computed the distribution of p53 residence times in unstimulated conditions and at 2 h and 4 h after irradiation with 10 Gy IR (Supplementary Movie 3). We found that in unstressed conditions the distribution of p53-binding times was well described by a bi-exponential decay, with an average residence time of $3.1 \pm 0.5$ s (Fig. 2b, c), in excellent agreement with previous estimates for live-cell p53 residence times obtained by ensemble average and single-molecule tracking approaches in unstressed cells[20–22]. Strikingly, the duration of the binding events increased transiently 2 h after the induction of damage and reverted back to basal levels 2 h later (Fig. 2b). On average, the duration of p53-binding events 2 h after the induction of genotoxic stress was twice as long as in basal conditions (Fig. 2c, upper panel). Combining this information on the average residence time and on the fraction of the bound molecules we could also estimate the average free time of p53 molecules between binding events (See Methods for details of the calculation), which remained unchanged before and after the induction of DNA damage (Fig. 2c, lower panel). Together these results indicate that the fraction of p53 bound molecules is determined by tuning the time p53 remains bound on chromatin, rather than by modulating the search process for these binding sites.

Interestingly, closer inspection of the bi-exponential fit of the distribution of residence times revealed that the only parameter that was tuned following p53 activation by IR was the residence time of the long-lived population of bound molecules (Fig. 2d). For p53 and other TFs, long-tails in the distribution of residence times have been interpreted as binding to specific response elements (REs)[23, 28, 29]. To prove this for our case, we performed single-molecule imaging on a mutant (HaloTag-p53mSB) which has been shown to be incapable of site-specific binding and transcriptional activation[23]. When stably expressed over a p53 null background (Supplementary Movies 4-7), this mutant p53 showed neither an increase in its bound fraction nor an extended tail of long-binding events (Supplementary Fig. 3), in stark contrast to what we observed with wild-type p53. These observations therefore support the hypothesis that p53-wt responds to genotoxic stress signals by modulating its residence time at specific REs on DNA—that is the residence time of the long-lived population of bound molecules.

We can combine the information obtained about non-specific and specific p53 binding on DNA to fully characterize the p53 search process for its target sites following its activation by IR (see Methods for the calculations), resulting in a search time of ~ 100 s.

**The increase in p53 residence time is due to acetylation of the p53-CTD.** We next focused on the possible causes underlying the modulation of p53 binding following activation by IR. We considered two main possible mechanisms: p53 binding might be enhanced by the increase in p53 levels due to cooperative binding, or p53 PTMs and in particular p53-CTD acetylation might render p53 binding to REs more stable, since we measured

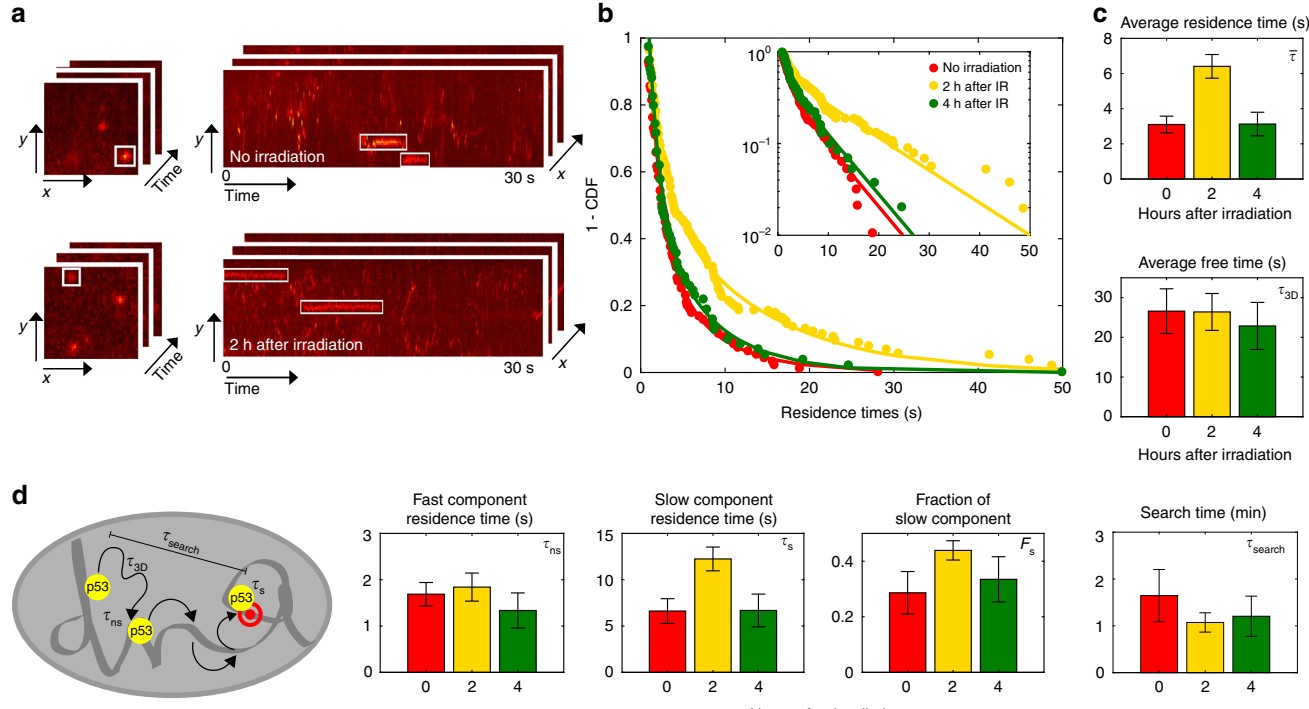

**Fig. 2** Measurement of the p53 residence time by kymograph analysis. **a** In kymograph representation, chromatin-bound transcription factors (*white rectangles*) appear as horizontal segments parallel to the temporal axis: the duration of each binding event is provided by the length of the segment. Typically, 2 h after irradiation p53 displays longer binding events than in unstimulated conditions. **b** The measured binding events are used to populate the complement cumulative distribution function (3 replicates, $n_{cells} = 18$, 23, and 20, $n_{bound} = 108$, 171, 81 for 0 h, 2 h, and 4 h, respectively) which is then corrected for the observational photobleaching and fit by a bi-exponential distribution (See Supplementary Methods). **c** The model provides estimates for the average residence time of p53 to chromatin and combined with the information on p53 bound fraction (Fig. 1) allows estimating the average free time between binding events. **d** The bi-exponential fit also provides estimates for the residence time of the short-lived population, for the long-lived residence time, and for the fraction of molecules in each of these states. The time that p53 spends searching for these stable sites is calculated as described in the Supplementary Methods (*error bars*: 95% CI)

p53 acetylation levels at Lysine 382 that correlate well with the observed modulation in p53-binding kinetics (more stable 2 h post IR than in basal conditions and 4 h post IR) (Fig. 3a).

We first verified that the increase in p53 levels observed following activation is per se not sufficient to induce enhanced binding to DNA: inducing the overexpression of HaloTag-p53 by treating our inducible cell line with doxycycline for 2 h resulted in no significant modulation of p53 bound fraction (Fig. 3b) or residence time (Fig. 3c), although the expression levels of HaloTag-p53 was similar to what observed 2 h post IR. Interestingly, we observed a modulation of p53 residence times when inducing p53 expression for longer times (Fig. 3b, c), and such modulation was accompanied by the acetylation of the p53-CTD at K382. We next tried to interfere with the acetylation state of p53 following DNA damage by exposing cells to Wortmannin, a PI3-Kinase inhibitor that prevents p53 acetylation at lysines K373 and K382[12]. By applying Wortmannin half an hour after the induction of DNA damage by IR, we were able to prevent p53 acetylation at K382 (Fig. 3d): in these conditions we measured a shorter p53 residence time at cognate-binding sites compared to the IR only case (Fig. 3e, f).

To directly connect the stability of p53 binding and the modification state of the p53-CTD, we next applied our SMT approach after interfering with Set8, a methyltransferase known to selectively modulate the methylation/acetylation state of the p53-CTD at K382[30] (Fig. 4a). Previous work has shown that silencing of Set8 results in the selective inhibition of K382 methylation that translated into an increased accessibility to K382 acetylation and to the activation of p53-mediated

transcription[12, 30]. While no significant modulation in p53 levels was observed upon silencing of Set8, we could appreciate an increase of the p53 bound fraction and of the average p53 residence time on chromatin (Fig. 4b), supporting the idea that p53 modifications at the CTD control the stability of p53 binding to REs.

Next, we performed SMT on p53 mutants mimicking or preventing acetylation of the CTD (Fig. 4a). First, we transiently transfected p53 null cells (H1299) with either HaloTag-p53wt, with a mutant mimicking acetylation at K382, HaloTag-p53-K382Q, or with another mutant, HaloTag-p53-6Q, mimicking acetylation of all the six lysines of the p53-CTD. With the exception of the 6Q mutant, all transfected plasmids displayed comparable expression levels (Fig. 4a). Transfected cells displayed faster dissociation of the TF from DNA than our stable cell lines expressing low levels of tagged p53, possibly due to the saturation of the most stable p53-binding sites. Nevertheless, the p53-K382Q mutant displayed an increase in p53 bound fraction and p53 residence time when compared to Halotag-p53-wt (Fig. 4c). Interestingly the p53-6Q mutant resulted in just a small further increase in p53 binding (Fig. 4c), suggesting that the acetylation of K382 residue is sufficient to stabilize the p53 interactions with DNA. Finally, we compared the single-molecule behavior of HaloTag-p53-K382R (a mutant where lysine 382 cannot be acetylated) before and 2 h after the activation stimulus provided by ionizing radiation. Differently from transiently transfected p53-wt, the K382R mutant did not display any modulation in p53 binding (Fig. 4d). We repeated this experiment in MCF-7 cells, after knocking out the expression of endogenous p53 by

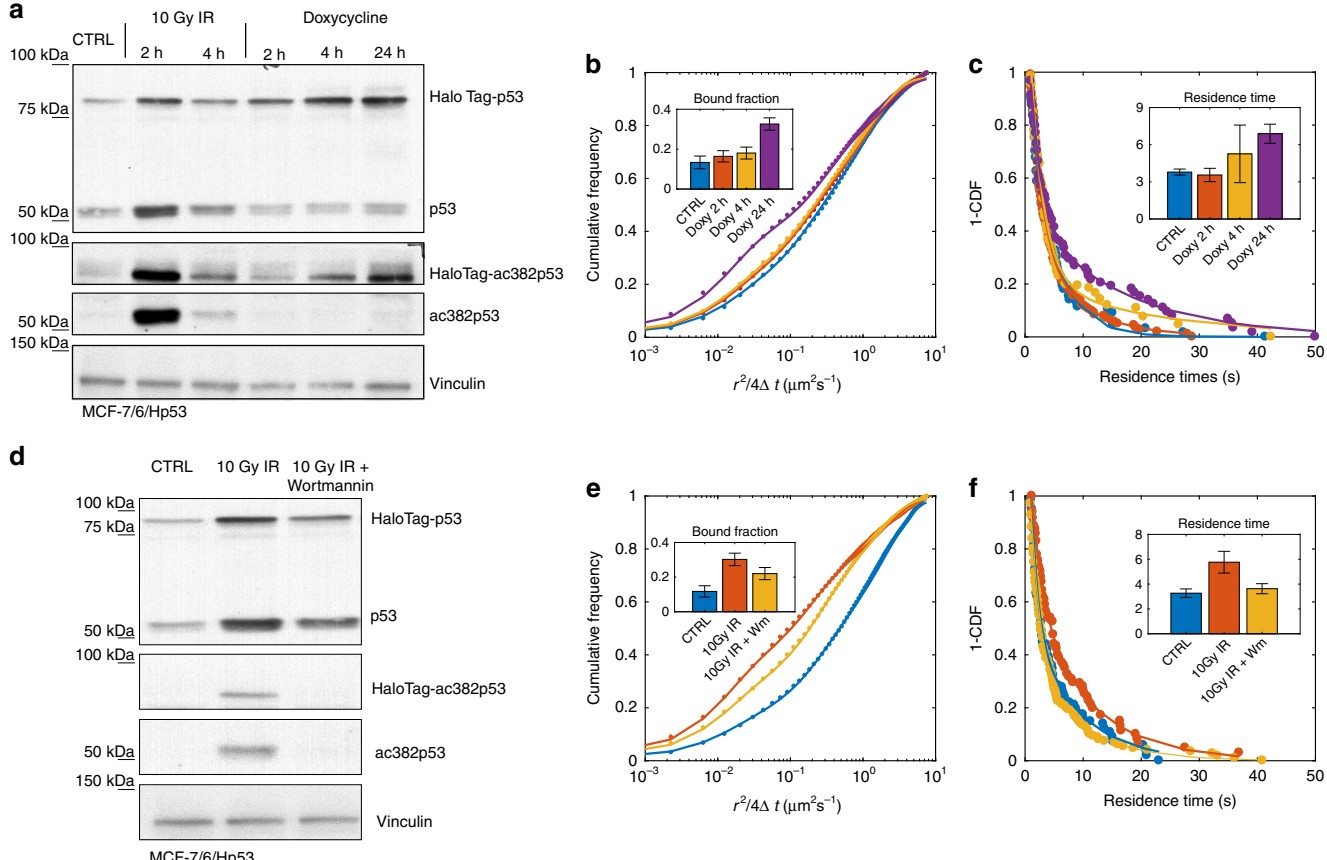

**Fig. 3** An increase in p53 levels is not sufficient in modulating p53-binding kinetics but p53-CTD acetylation is also necessary. **a** Western blot of HaloTag-p53 total protein and acetylated at lysine 382 upon exposure to 10 Gy IR or Doxycycline. Quantifications of western blots are provided in Supplementary Fig. 4. **b** Complement cumulative distribution of single-molecule displacements at different times upon induction of HaloTag-p53 expression by doxycycline and measured bound fraction (*inset*, 3 replicates, $n_{cells}$ = 25, 19, 25, 9, $n_{displacements}$ = 14 278, 14 420, 10 807, 11 227 for 0 h, 2 h, 4 h, and 24 h after doxycycline, respectively. **c** Distribution of residence times and average residence times (*inset*) following doxycycline induction ($n_{bound}$ = 80, 180, 67, 80). **d** Western blot of HaloTag-p53 total protein and acetylated at lysine 382 upon irradiation and upon the combination of IR and Wortmannin. Quantifications of western blots are provided in Supplementary Fig. 4. **e** Cumulative distribution of displacements (2 replicates, $n_{cells}$ = 17, 15, 15, $n_{displacements}$ = 17 592, 9846, 17 272 for Ctrl, IR, IR+Wort, respectively) and **f** the complement cumulative distribution of residence times ($n_{bound}$ = 115, 107, 172) upon irradiation and treatment with Wortmannin (*error bars*: SD for bound fractions, 95% CI for average residence times)

CRISPR/Cas9 and re-inserting HaloTagged p53-wt or p53-K382R by transient transfection (Supplementary Fig. 5). The data confirmed that acetylation of p53-CTD is necessary for the stabilization of p53 binding. Further, the measured bound fractions were found to be weakly negatively correlated to the levels of p53 expression in single cells, confirming that the increase in p53 levels is not responsible for the increase of p53 binding following the activation by DNA damage.

Taken together, these data indicate that p53 acetylation at the CTD, and in particular at K382, is both a necessary and a sufficient condition to induce the stabilization of p53 binding, and therefore that the p53-binding kinetics can be considered a physical read-out of the acetylation state of the p53-CTD.

**p53 acetylation and binding correlate with transcription.** We next tested the capability of p53 to transactivate target genes in living cells depending on its binding kinetics and the acetylation state of the CTD. To this end, we compared transcription of known p53 target genes by quantitative real-time PCR assay (qPCR) at different times after the induction of DNA damage (where both p53 binding and CTD acetylation are transiently modulated), and after the induction of HaloTag-p53 over-expression by doxycycline (where the rapid increase of p53 levels

accompanied by a slower stabilization of p53 binding and CTD acetylation). Transcription of the tested genes was found to rapidly increase following DNA damage (Fig. 5a), but with slower kinetics and to a lesser extent following doxycycline induction (Fig. 5b).

We further investigated the role of the TF expression levels in the transactivation of a canonical p53 target gene, *CDKN1a*, at the individual cell level, using single-molecule fluorescence in situ hybridization (smFISH)[31, 32]. We selected 20mer probes that targeted *CDKN1a* exons in order to detect and count cell by cell both mature RNA—which appears as individual foci scattered throughout the cell-RNA and nascent-RNA, which localizes at brighter foci at active transcription sites in the nucleus[32–35] (Fig. 5c). The quantification of nascent transcription by smFISH correlated well with population measurements by qPCR (Supplementary Fig. 6a–b) and confirmed that transcription of *CDKN1a* upon irradiation was p53-dependent (Supplementary Fig. 6c).

As expected, cells not exposed to damage displayed significantly lower amounts of both nascent and mature RNA than irradiated cells (Fig. 5c, d). Importantly, even cells with high nuclear HaloTag-p53 levels in non-damaged conditions displayed lower amounts of nascent *CDKN1a* transcript than damaged cells with similar levels of HaloTag-p53 (Fig. 5e). Further, the number of nascent *CDKN1a* transcripts was uncorrelated to the levels of

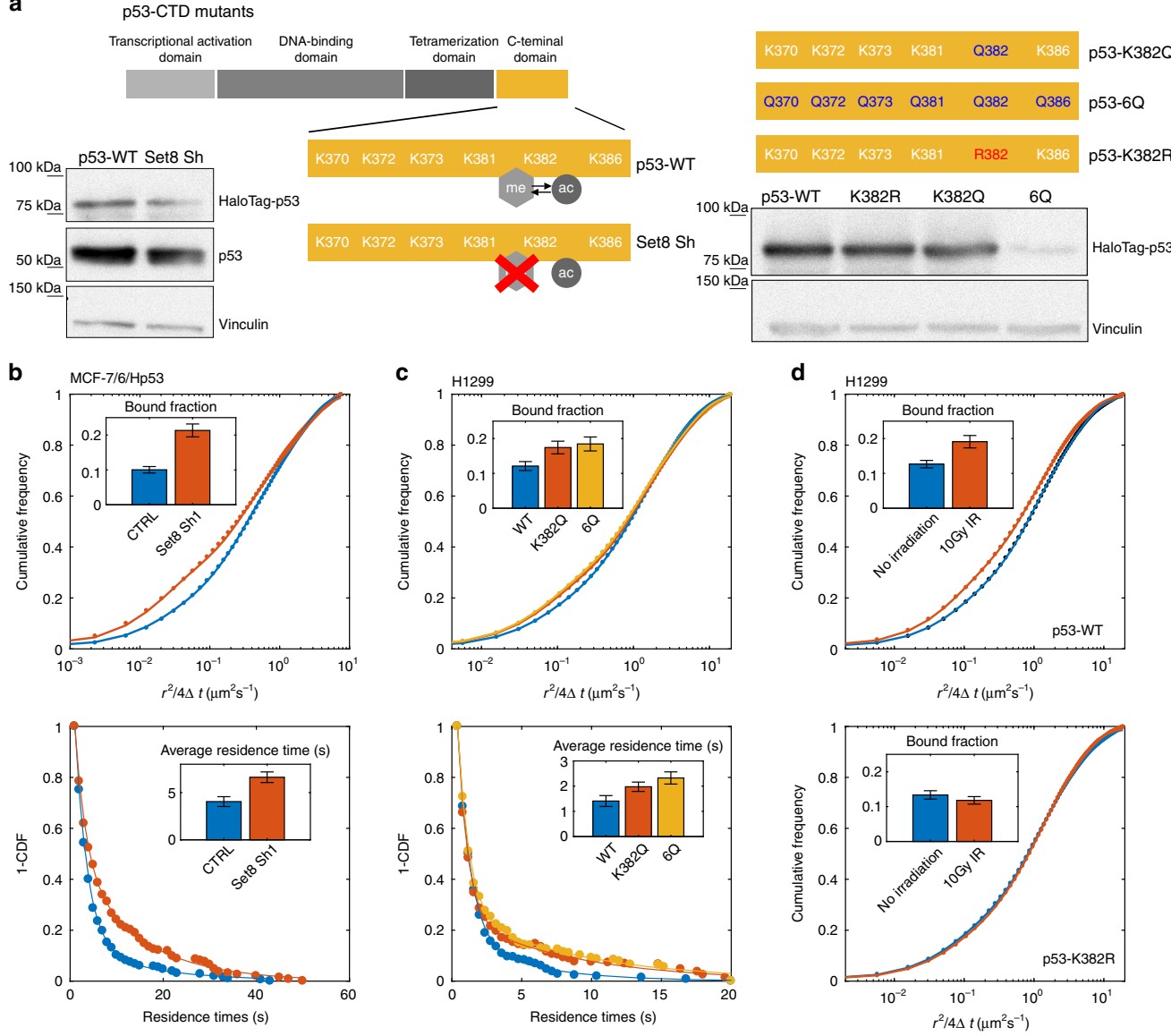

**Fig. 4** Dissecting the role of CTD acetylation in binding. **a** Scheme of the treatments and p53 mutants. Expression levels of (HaloTag)-p53 upon the different treatments are shown. **b** Cumulative distribution of displacements (*top*) and complement cumulative distribution function of the single-molecule residence times (*bottom*) for control MCF-7/6/Hp53 cells, and cells transfected with a short-hairpin inhibiting the methyltransferase Set8 (3 replicates, $n_{cells} = 22$, 23 for CTRL and Set8 Sh1, respectively). **c** Distribution of displacements (*top*) and complement cumulative distribution function of the single-molecule residence times (*bottom*) for HaloTag-p53-wt, HaloTag-p53-K382Q, or HaloTag-p53-6Q (acetylation mimicking mutants) transiently transfected in H1299 cells (*insets*, 3 replicates, $n_{cells} = 28$, 24, and 18 for WT, K382Q, and 6Q, respectively). **d** Distribution of displacements for HaloTag-p53-wt (*top*, 2 replicates, $n_{cells} = 19$ for no irradiation and 16 for 10 Gy IR) and p53-K382R (acetylation inhibiting mutant, *bottom*, 2 replicates, $n_{cells} = 16$ for no irradiation and 15 for 10 Gy IR) transiently transfected in H1299 cells before and 2 h after irradiation (*error bars*: SD for bound fractions, 95% CI for average residence times)

HaloTag-p53 in non-damaged conditions. These results provide supporting evidence at the single cell level to previous assays demonstrating that p53 concentration does not determine transcription levels[12]. Importantly, we found that after the activation by genotoxic stress, the amount of nascent RNA was correlated to the p53 expression levels only at specific time points (2 h after the induction of DNA damage) but not at later time points such as 4 h post IR. i.e., only at those time points in which p53 was found to be acetylated and to bind more tightly REs on DNA (Fig. 5e). Interestingly, we also found that individual active transcription sites displayed, on average, the same number of nascent transcripts (Fig. 5f, right panel) at 0 h, 2 h, or 4 h post IR. This suggests that the modulation of *CDKN1a* nascent

transcription might occur mostly by tuning the duration of transcription bursts (Fig. 5f, left panel), with every burst possibly saturating the transcription unit with the maximal number of PolII eleongating complexes that can be accommodated.

These results lead us to propose a model in which the more stable binding of p53 for its targets can act as a clutch coupling p53 abundance to the capability of transactivating target genes. To provide supporting evidence for our model for the *CDKN1a* target gene, we performed smFISH experiments upon expression of the p53 mutant which cannot be acetylated at Lysine 382, and found no increase in *CDKN1a* transcription following irradiation (Supplementary Fig. 6d). Finally, we replicated our smFISH analysis to the other conditions used throughout this work (Ectopic HaloTag-p53

expression by the induction with doxycycline and inhibition of acetylation by Wortmannin, Fig. 5g) and found that the average number of *CDKN1a* nascent transcripts correlated better with the p53 acetylation state and with the average p53 residence time rather than with the average p53 amount (Fig. 5g and Table 2).

## Discussion

The activity of many TFs regulating inducible gene expression, such as p53, steroid receptors, and NF-kB, is modulated by

controlling their concentration in the nucleus[3, 36]. This mechanism is often excitable (i.e. irreversible once started) and therefore provides a rapid and effective method to respond promptly to external cues[11, 36]. Excitable activation of a TF has however its downsides, as intrinsic fluctuations typical of the cellular noisy environment might inadvertently activate the full TF response. This might be particularly dangerous for p53, as the activation by a transient spontaneous fluctuation could lead to death by apoptosis of a healthy cell. Additional regulation layers,

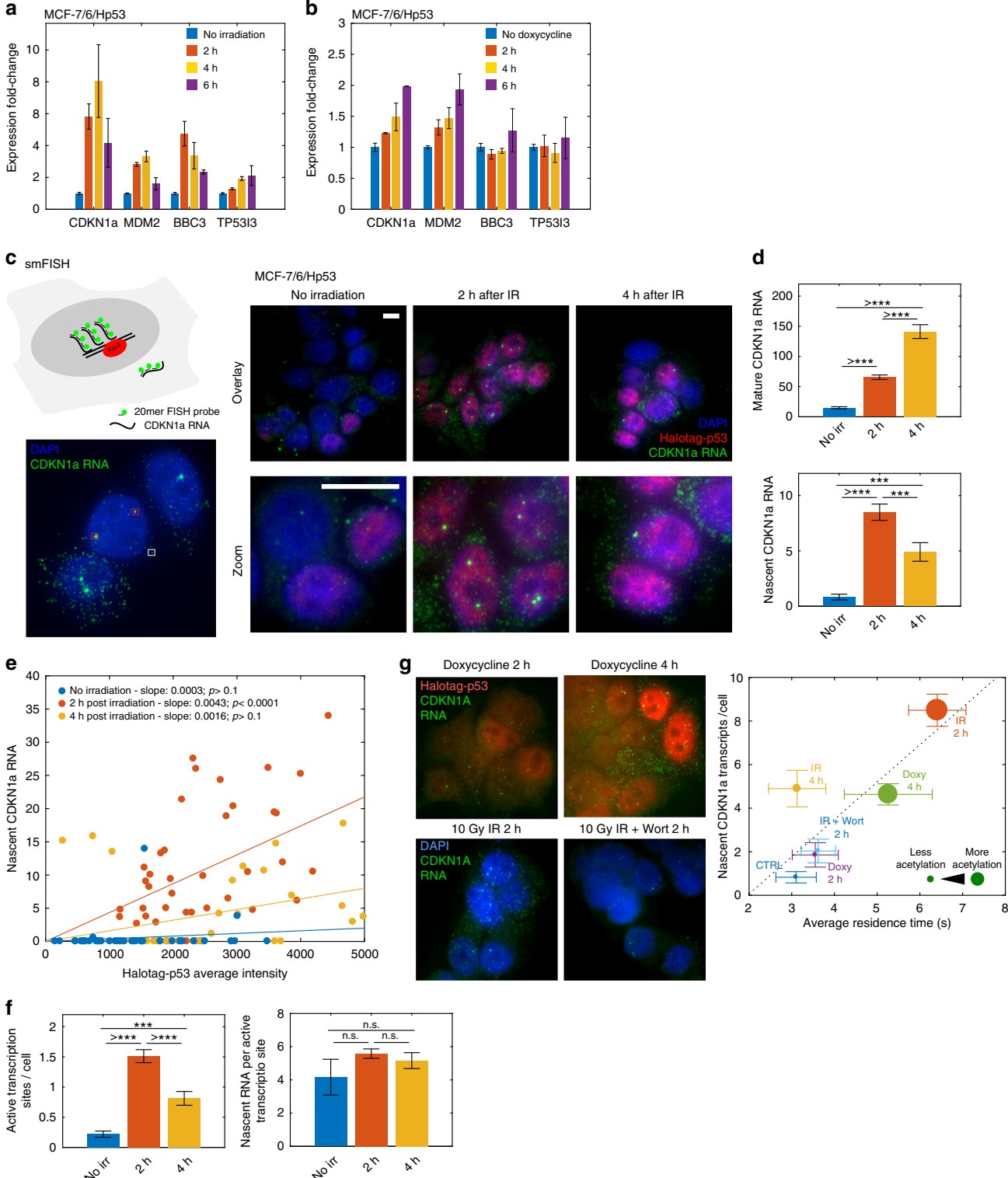

mediated by reversible PTMs, have therefore been proposed to achieve the fine balance between mounting a rapid and robust response to external sources of stress and filtering out signals arising from cellular noise[37, 38]. It has been shown that C-terminal modifications of p53 can modulate its transcriptional activity[39], but the role of this modifications in tuning the p53 affinity to cognate sites in living cells vs the increase of p53 nuclear levels in living cells following activation by stress signals has remained elusive[16, 40]. Here we applied single-molecule imaging approaches and proved that p53-binding kinetics to its target sequences on DNA is indeed modulated by C-terminal acetylation upon activation by DNA damaging agents such as IR.

Our single-molecule tracking experiments reveal that in both basal and activated conditions p53 scans the nuclear environment by a combination of three-dimensional diffusion and short-lived (few seconds) immobilization events on chromatin. As a result p53 spends most of its time free, a common feature of multiple eukaryotic TFs[26, 28, 41], which might be due to inefficient association to chromatin caused by competition with other DNA-binding proteins or nucleosomes[42, 43]. Our measurement of p53-binding kinetics indicates that the dwell time of p53 when non-specifically bound to DNA is about one order of magnitude shorter than the dwell time at specific sites ($\sim$1 s vs $\sim$10 s). Interestingly, in vitro p53 affinities measured for DNA either containing or lacking specific-binding sites display comparable differences[44]. Such a relatively small difference in affinities can be sufficient to warrant specific activation of target genes, only if p53 is allowed to scan hundred-basepair stretches of DNA by 1D sliding or to hop intersegmentally while non-specifically bound[42]. Indeed, in vitro single-molecule experiments have confirmed that p53 can slide on DNA[45] and live-cell SMT data support this model[22].

Our SMT data provide direct evidence that the fraction of p53 molecules engaged in chromatin binding is modulated following activation, resulting in a transient increase in the first 2 h after IR. The modulation in the quota of bound p53 does not appear to be achieved by changes in the p53 search mechanism, as the average free time spent by p53 molecules varies only slightly before and after the induction DNA damage by IR, in agreement with recent genome-wide data showing that most of the p53-REs are pre-bound in unstressed conditions[40, 46, 47]. Rather, our data imply that genotoxic stress results in the stabilization of p53 binding at putative specific-binding sites. While remaining transient, the p53 residence time at these sites doubles from about 5 s in basal conditions to about 12 s upon activation. We were able to pinpoint acetylation of the CTD as the biochemical event underlying the modulation of p53 residence time.

Further, by combining SMT experiments with measurements of RNA synthesis, we have shown that the activation of p53 by IR results in a transient increase in transcription of a number of target RNAs including *CDKN1a*. The increase in transcription cannot be solely attributed to the increase in p53 levels, but rather correlates with the stability of p53 binding. Similar correlations have been shown to hold for other TFs at artificial arrays of binding sites[48] and lead us to propose an "affinity/abundance clutch" model where the increase in both the number of p53 molecules and in the binding stability of post-translationally modified p53 molecules plays a role in determining the transcriptional response to stress signals. A similar model has been proposed from genome-wide competition ChIP experiments[7], showing that the strength of binding to REs correlates with the transactivation of downstream genes. As the proposed model connecting TF-binding kinetics and transcriptional activation is supported by correlative evidence, future experiments—potentially based on measuring TF kinetics at individual REs, combined with genome editing—will be needed to identify a causative link between these two variables.

These experiments might prove particularly challenging, given the difficulties of labeling endogenous genetic loci, without altering their response to activating stimuli, but a potential solution might be represented by CRISPR/Cas9-mediated genetic labeling[49].

At this stage, we can only speculate on how the modulation of p53 residence time could affect the transcriptional outcome. One possibility is that by binding longer on REs p53 would increase the probability of recruiting co-factors necessary for activating a round of transcription, while higher p53 levels might increase the frequency at which these more stable binding events would occur at a specific transcriptional locus. Accordingly, we have recently shown that sites that show the tighter p53/DNA binding also display an accumulation of polymerase II[23].

Our data show that the p53 residence time correlates with the number of active transcription sites per cell. This might indicate that p53 residence time can tune the duration of transcription bursts, in agreement with recent smFISH studies on other TFs[35].

Our finding that the number of nascent transcripts present at each transcription site is invariant among the different tested condition indeed suggests that when the gene is on it actually produces transcripts at a predetermined and constant rate. According to this model, the transcriptional output would depend on the probability of the switching on of the gene—which will depend on the TF abundancy—and the on the duration of the transcriptional burst (or, equivalently, the probability of switching off from the on state), which might depend on the TF residence time. In this scenario, some of the co-factors in pre-initiation complex formation would be capable of translating the differences in p53 residence times into an "on/off" switch, possibly by reading the acetylation state of the p53-CTD. The methods described in this work will allow testing the possibility that either the residence time and/or the search mechanism of p53-recruited co-factors are affected by the stability of p53 binding to chromatin.

**Fig. 5** Population and single-cell transcriptional response to the modulation of p53-binding kinetics. **a** qPCR of p53 targets following irradiation (3 replicates, *error bars*: SD). **b** qPCR of p53 targets following induction of HaloTag-p53 expression by doxycycline (3 replicates, *error bars*: SD). **c–g** smFISH imaging of *CDKN1a*. **c** smFISH is performed by hybridizing multiple labeled oligonucleotides to the specific RNA and acquiring 3D stacks to count mature RNAs (*white square*) and nascent RNAs at transcription sites (*red square*, maximum projection displayed). **d** Average amount of mature (*top panel*) and nascent (*bottom panel*) RNA per cell (*top panel*, 2 replicates, $n_{cells}$ = 91, 121, 64 for 0 h, 2 h and 4 h, respectively; ANOVA-Tukey test). The measurement of nascent *CDKN1a* RNA was validated by qPCR, using primers targeting pre-spliced *CDKN1a* RNA (See Supplementary Fig. 4) **e** Correlation of cell-by-cell number of nascent transcripts vs. nuclear intensity of HaloTag-p53. Nascent *CDKN1a* is correlated with p53 levels 2 h after the induction of damage (Pearson correlation, $r = 0.55$, $p < 0.0001$, slope = 0.0043) but not before the induction of damage ($r = 0.12$, $p > 0.1$), nor 4 h after ($r = 0.27$, $p > 0.1$), **f** Number of detected active *CDKN1a* transcription sites per cell (*left panel*) and amount of nascent *CDKN1a* RNA per active site (*right panel*) (2 replicates, $n_{cells}$ = 91, 121, 64 for 0 h, 2 h, and 4 h, respectively; ANOVA-Tukey test). **g** Exemplary smFISH for the treatments with Doxycycline and IR+Wortmannin (*top-left*). Correlation between p53 residence time, number of active *CDKN1a* transcription sites per cell and p53-CTD acetylation across the tested conditions (*top-right*). The radius of the displayed dots is proportional to the amount of HaloTag-p53-Ac382. Correlations coefficients are shown in Table 2 (*error bars*: SEM for FISH data, 95% CI for residence times)

**Table 2 Correlations between p53 residence time, number of active CDKN1a transcription sites per cell and, p53 levels and p53-CTD acetylation**

| Pearson's correlations | | |
| --- | --- | --- |
| P53 levels vs. nascent CDKN1a | $\rho = 0.228$ | $p > 0.5$ |
| P53 acetylation vs. nascent CDKN1a | $\rho = 0.864$ | $p < 0.05$ |
| P53 residence time vs. nascent CDKN1a | $\rho = 0.821$ | $p < 0.05$ |
| P53 residence time vs. p53 acetylation | $\rho = 0.93$ | $p < 0.01$ |

CDKN1a transcription better correlates with p53 residence time and p53-CTD acetylation than with p53 abundancy

Measurement of TF kinetics at individual REs might also allow to check whether the modulation of p53-binding kinetics occurs at all p53-dependent promoters, as at present we can only claim that on average residence times on chromatin increase at some number of p53 REs. Nevertheless, our data on transcription of p53 targets seem to indicate that all of the tested genes are dependent on the acetylation state of p53, although to different extents. Targets such as *CDKN1a* or *PUMA* show the largest difference in transcription induction depending on the p53 acetylation state while others such as *MDM2* appear to be less dependent on p53 acetylation, in agreement with the different dependence on p53 acetylation of in vitro p53 affinities that have been reported for the *CDKN1a* and the *MDM2* promoters[50]. The *MDM2* promoter might have evolved to be particularly sensitive to the frequencies of p53-binding events, which is directly proportional to p53 abundance. Possibly, in this case, the clutch may be set to "self-drive".

In sum, we have shown that the kinetics of transient p53/chromatin interactions can be modulated upon activation and that the stability of the binding reflects a modulation of the p53-CTD acetylation state. The methods described here can now be applied to generalize the effect of modifications to TF-binding kinetics and its correlation with transcriptional activity to other TFs and co-factors.

## Methods

**Plasmids and transient transfections.** Plasmids encoding for CTD mutations of HaloTag-p53 were obtained from the CMVd1-HaloTag-p53wt by site directed mutagenesis by GeneWiz (GeneWiz, South Plainfield, NJ, USA). The entire plasmid sequences were verified prior to use. H1299 cells were transiently transfected with these plasmids using the JetPrime transfection reagent (Polyplus-transfection, New York, NY, USA) and labeled and imaged 18 h after transfection.

The plasmid for the expression of Set8 short-hairpin was a kind gift from Gailit Lahav[12]. MCF-7/6/Hp53 cells were transiently transfected with the Set8-sh plasmid, kept under antibiotic selection for 72 h and then labeled and imaged as described below.

**Stable cell lines generation.** The plasmids encoding for HaloTag-p53wt and HaloTag-p53mSB under the control of a Tet-dependent promoter were generated by digestion of the CMVd1-HaloTag-p53 and CMVd1-HaloTag-p53mSB plasmids[23] with HindIII and XbaI restriction enzymes, and subsequent ligation into the pCDNA-4/TO vector (Invitrogen, Thermo-Fisher, Waltham, MA, USA). Each of the plasmids was co-transfected with the pCDNA-6 plasmid (Thermo-Fisher) (responsible for the constitutive expression of the Tet-repressor) into lung carcinoma H1299 cells (ATCC, LGC Standards S.r.l., Milan, Italy) using Lipofectamine LTX (Thermo-Fisher) according to the manufacturer's instructions. Breast carcinoma MCF-7 cells (ATCC) were also transfected with pCDNA-4/TO/ HaloTag-p53wt and pCDNA-6 using Lipofectamine LTX. Cells expressing Tet-repressor and HaloTag-p53 were selected by antibiotic resistance using 10 μg ml⁻¹ Blasticidin and 150 μg ml⁻¹ Zeocin (Thermo-Fisher). Individual clones were generated by serial dilution and screened for expression levels, nuclear localization of the tagged protein and for its capability to induce transactivation of target genes (Supplementary Fig. 1). Cells were routinely tested for mycoplasma contamination by qPCR.

**Generation of MCF-7-p53KO Cells.** MCF-7 cells were transfected using Ultra-Cruz Transfection Reagent (sc-395739) with constructs expressing Cas9-D10A (Nickase) and sgRNAs targeting p53 exon3 (Santa Cruz sc-416469-NIC). 24 h after transfection, cells were selected in culture medium containing 1 μg ml⁻¹ puromycin for 5 days. Then, cells were suspended and seeded to allow single clone formation, in complete medium without antibiotic. Picked clones were analyzed for the expression of p53 both by western blot and by immunofluorescence with α-p53 (DO-1) antibody (Supplementary Fig. 5a).

**Cell culture.** MCF-7 and H1299 cell lines were cultured in RPMI-1640 medium (Gibco, Thermo-Fisher, Waltham, MA, USA) supplemented with 10% heat-inactivated Fetal Bovine Serum, 2 mM L-Glutamine, 100 units ml⁻¹ penicillin and 100 μg ml⁻¹ streptomycin.

HaloTag-p53 levels and localization in MCF-7/6/Hp53 cells were assessed by incubating with 500 nM HaloTag-TMR for 30 min followed by extensive washes with PBS to remove the unbound fluorescent ligand. Cells were fixed in 4% paraformaldehyde for 10 min at room temperature (RT), incubated with 1 μg ml⁻¹ Hoechst 33342 (Invitrogen) for 10 min at RT and extensively washed in PBS.

**Immunofluorescence.** Immunofluorescence for p53 in MCF-7 cells was carried out by fixing cells with 4% paraformaldehyde for 10 min at RT, permeabilizing them with 0.1% TritonX-100, blocking with 5% BSA in PBS and incubating them with mouse monoclonal p53 antibody [DO-1] (Abcam Ab1101, Cambridge, UK) diluted 1:200 in PBS-0.1% Tween20-2% BSA for 2 h at RT. Cells were washed three times with PBS-0.1% Tween20-2% BSA and incubated with antimouse AlexaFluor 647 secondary antibody in PBS-0.1% Tween20. Cells were washed for three times with PBS and the DNA was labeled with 1 μg ml⁻¹ Hoechst 33342 in PBS 10 min at RT. Imaging of the mounted coverslips was performed by using a confocal microscope (TCS SP5 AOBS Leica LSM, Leica Microsystems SRL, Milan, IT) equipped with a 63×NA1.4 oil-immersion objective.

**Irradiation and drug treatment.** In order to induce activation of p53 by genotoxic stress we irradiated MCF-7 and H1299 cells with 10 Gy gamma-ray irradiation using a ¹³⁷Cs source (Biobeam 2000).

In order to prevent p53-CTD acetylation we used 100 μM wortmannin (Sigma Aldrich SRL, Milan, Italy), added half an hour after the induction of DNA damage. The wortmannin solution was replaced fresh every hour. All the experiments on HaloTag-p53 were performed by exploiting the leakage of the Tet-regulated promoter, except when HaloTag-p53 overexpression was desired, where concentrations of doxycycline ranging from 1 ng ml⁻¹ to 1 μg ml⁻¹ were used. DMSO was added in control samples.

**Western blotting.** Cells grown on 10 cm plates were washed once in PBS and lysed in 500 μl Lysis Buffer (50 mM Tris HCl pH 7.5, 150 mM NaCl, 1% NP40, 5 mM EDTA) with protease inhibitors (Sigma-Aldrich, Milan, Italy). The samples were incubated at 4 °C, for 20 min and centrifuged at 15 000 rcf for 15 min to collect the supernatant. An aliquot of the cell lysate was used for protein quantification with BCA (Thermo-Fisher). Proteins were separated in 10% SDS-polyacrylamide gels and transferred to nitrocellulose membranes, in Transfer Buffer (25 mM Tris, 192 mM Glycine, 20% Methanol) at 100 Volts for 1 h. Membranes were blocked using 5% non-fat dried milk in TBS-T (0.1% (v/v) Tween20 in 1× TBS) for 1 h at RT. The antibodies used for immunoblotting were: mouse monoclonal anti-p53 DO-1; rabbit monoclonal anti-p53 acetyl K382 (Abcam 75754, diluted 1:1000 and incubated at 4 °C overnight); mouse monoclonal anti-vinculin (Millipore CP74, diluted 1:10 000 and incubated for 1 h at RT); mouse monoclonal anti-GAPDH (Sigma-Aldrich G8795; diluted 1:2000 and incubated for 1 h at RT). Peroxidase-conjugated secondary antibody anti-mouse IgG (Cell Signaling, 7076) or anti-rabbit IgG (Cell Signaling, 7074) were diluted 1:5000 and hybridized for 1 h at RT. All the antibodies were diluted in TBS-T with 5% non-fat dried milk. Full scans of the western blots shown in Figs. 1 and 3 are provided in Supplementary Fig. 7.

**RNA extraction and quantitative real-time PCR assay.** RNAs were extracted using TRIzol Reagent (Invitrogen) and subsequently purified using silica membrane spin columns from Nucleospin RNA (Mechery Nagel). RNA quantity and purity were assessed using a NanoDrop fluorimeter (Thermo-Fisher). 4 μg of total RNA were reverse-transcribed using the random hexamers-based High Capacity cDNA Reverse-Transcription Kit (Applied Biosystem), according to the manufacturer's instructions.

Gene expression of p53 targets was measured using qPCR. The reaction was performed in a final volume of 20 μl, using the LightCycler 480 SYBR Green I Master mix (Roche, Monza, IT), 0.5 μl of diluted cDNA (1:100), and 0.5 nM of each primer. Normalization of cDNA loading was obtained by running all samples in parallel using human GAPDH as a loading control. The cDNA was amplified using an initial denaturation and activation step at 95 °C for 10 min, followed by 40 cycles of 95 °C for 15 s, 60 °C for 1 min, and 95 °C for 15 s. The sequences of the primers used are provided in Supplementary Table 1.

**Single-molecule fluorescence in situ hybridization.** To identify *CDKN1a* nascent and mature RNA, Design Ready Stellaris RNA FISH probes (Biosearch Technologies, Petaluma, CA, USA), labeled with the Quasar 570 fluorescent dye were used according to the manufacturer instructions. Cells were fixed in 4% PFA

for 10 min at RT, followed by washing in PBS and permeabilization with 0.1% TritonX-100. Cells were then incubated for 10 min at RT with 1 ml of washing buffer A composed by 10% saline sodium citrate (SSC), 10% formamide solution (Sigma Aldrich), diluted in RNase-free water. Hybridization was carried out overnight in a humidified chamber at 37 °C by covering the coverslip with 0.5 μl of 12 μM *CDKN1a* FISH probes diluted in 100 μl of hybridization buffer, composed by 10% (w/v) of dextran sulfate, 10% of SSC-20× buffer and 20% formamide in Rnase-free water. After the hybridization, cells were washed twice in buffer A for 30 min at 37 °C, washed once in 10% SCC-20×. Following DNA staining with 1 μg ml−1 Hoechst 33342 in PBS, the coverslips were mounted on glass slides using Vectashield (Vector Laboratories, Peterborough, UK) mounting media. To monitor HaloTag-p53 levels together with *CDKN1a* transcription, cells were labeled with 5 μM HaloTag-SIR fluorescent ligand[51] for 30 min at 37 °C before cell permeabilization. Cells were then extensively washed to remove the unbound ligand.

Imaging was performed on our single-molecule capable microscope (see below), by using the mercury lamp as excitation source. For every field a z-stack series of images were acquired with 0.3 μm step size, to count the total number of mature RNAs for each cell.

smFISH stacks were analyzed using a previously described Matlab (Matworks, Natick, MA) analysis package, named FISHquant[34]. Mature RNAs were identified as 3D gaussian spots with intensity higher than an arbitrary threshold, which was held constant for all acquisitions belonging to the same experiment. Nascent RNAs at active transcription sites were quantified by identifying the sites of nascent transcription as bright (>10 times higher than the threshold set for mature RNA), nuclear foci. With these settings, no more than four actively transcribed loci were found within each nucleus. For each of the transcription sites the amount of RNA was calculated by comparing the integrated intensity of the site with the average integrated intensity of the spots identified as mature RNAs. The nascent RNA measurements by smFISH were validated by qPCR (Supplementary Fig. 4). For visualization, the smFISH stacks were deconvolved using the Iterative Deconvolve plug-in of FiJi[52] and maximally projected.

**Single-molecule tracking**. *SMT microscopy*. For single-molecule imaging we used a custom-built microscope capable of inclined illumination[25], based on a Olympus IX-81 microscope frame, equipped with a ×100, 1.49 NA oil-immersion objective (Olympus Life science, Segrate, IT) and an EM-CCD camera (Photometrics Evolve 512, Photometrics, Tucson, AZ, USA), for a resulting pixel size of 145 nm. In order to obtain short (5 ms) exposures independently of the frame-rate used we adopted stroboscopic illumination, which was achieved by synchronizing the camera exposure signal to the digital modulation input of the 561 nm laser (iFlex Mustang 561 nm, QiOptiq Photonics GmbH, Munich, DE) through a pulse generator (Berkeley Nucleonics, San Rafael, CA, USA). All acquisitions were performed at 37 °C and 5% $CO_2$ by making use of an environmental chamber (Okolab SRL, Naples, IT).

*Cell labeling and acquisition*. The day before single-molecule tracking experiments $2 \times 10^5$ cells were plated on 2-well LabTek coverglass chambers. One hour before imaging HaloTag-p53 was labeled by using 500 pM Halotag ligand, incubated for 30 min at 37 °C and extensively washed (two rounds of three washes in PBS and 15 min incubation at 37 °C in phenol-red free RPMI). For each cell, we acquired a reference low-intensity fluorescence image and a transmitted light image, followed by the SMT acquisition, a time lapse of 500 consecutive images obtained by exposing for 5 ms with ~ 1 kW cm−2 of the 561 nm laser light, with a frame-rate ranging from 10 to 25 fps. All experiments were repeated at least twice on different experimental days.

*Analysis of the single-molecule tracking movies*. The collected SMT movies were analyzed by custom-written Matlab routines, to identify and track individual molecules, as previously described[22].

The resulting tracks were analyzed to quantify the p53 bound fraction as described in the Supplementary Methods. Single-molecule movies were also analyzed to quantify the distribution of p53 residence time on chromatin, as described in the Supplementary Methods. Tracking failed for cells with high density of molecules (approximately > 20 molecules/frame): these cells were excluded from further analysis.

**Data availability**. The data used to generate the plots displayed in the article have been deposited on Figshare (doi: 10.6084/m9.figshare.5119759). All other relevant data is available from the authors upon request.

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

## Acknowledgements

We are grateful to M. Panattoni and R. Pardi for providing the pCDNA-4/TO and pcDNA-6 plasmids, to L. Gianni for the p53-Ac328 antibody, to G. Lahav and J. Reyes for the plasmid encoding for the Set8 short-hairpin, to D. Gabellini and G. Tonon for critical discussions, to F. De Marchis for technical support, to F. d'Adda di Fagagna and to J.G. Mcnally for critical reading of the manuscript. The work was supported by the Marie Curie International Incoming Program (DM, GA-2010-274323) and by Fondazione Cariplo (A.L. and D.M.: 2014-1157). M.E.B. is supported by the Epigen flagship project.

## Author contributions

A.L., E.J., P.R., T.M., S.A., and D.M. generated and validated biological material and analysis routines. A.L., E.J., T.D., and D.M. performed the experiments. E.J., A.L., D.M. analyzed the experiments. M.E.B., C.T., and D.M. designed research. D.M. wrote the manuscript. All authors edited the manuscript.

## Additional information

**Competing interests:** The authors declare no competing financial interests.

