## [Peer Review File · Nature Communications]

Reviewers' Comments:

Reviewer #1 (Remarks to the Author):

A longstanding question in the p53 field revolves around how specific post-translational modifications govern the transcriptional output at the molecular level, which enables p53 to govern the downstream cellular response accordingly. In this paper, Jacchetti et al describes a novel molecular mechanism that links acetylation of p53, a known post-translational modification to p53 following irradiation, to the increased resident time of p53 on DNA and target gene promoters. Should this observation hold true, it does provide new insights into the roles of acetylation in regulating the promoter binding by transcription factors. However, as it stands, the results included are insufficient to enable the authors to draw the overarching conclusions, as several key experiments and controls are missing.

1. Inconclusive evidence to support that p53 levels alone does not alter bound fraction of p53:

The authors conclude that p53 level alone does not cause a change in bound fraction of p53 (Line 228-230), which would be reasonable based on the effects of doxycycline induced only at 2 hours (Fig 3a-c). However this conclusion is difficult to draw for 4 hours of Doxycycline addition as the total p53 has changed drastically. Furthermore, at the 24 hours Doxycycline addition time point, where a large change in p53 bound fraction is observed, there is no Western blot to accompany the graph to conclude whether it is a change in p53 level or acetylation that is causing the difference, or indeed other factors/reasons, such as activation of proteins that in turn modulate p53 behavior. Using Wortmannin as an indirect drug to inhibit K382 acetylation, the authors claim that the reduced acetylation at K382 (Line 236-239, Fig 3d-f) led to reduced p53 bound fraction. However in this case, although the Western blot shows a substantial reduction in K382-acetylation, the total level of p53 is equally reduced, therefore the alteration in p53 behavior may also arise as a result of p53 level, and not just due to acetylation. The same argument applies to Fig.4, where although the authors elegantly use Set8 knock down as well as acetylation mimic/blocking (KQ/KR) mutants to show the effect of acetylation on p53 bound fraction, the total level of p53 was not quantified. In order to rule out the potential effect of p53 levels, a Western blot is necessary to show that the total p53 level (regardless of its acetylation status/mutation) is unaltered.

2. Inconclusive evidence to support that promoter affinity does not alter resident time of p53:

This is an important point given that the authors have overexpressed Halo-tagged p53 in the background of endogenous p53, which poses a further complication on p53 level analysis. In all Western blots, the authors have only shown the Halo-tagged p53, but not the endogenous p53 in cells. It is possible that both the level and acetylation status of endogenous p53 levels have changed too. One can assume that should the Halo-tag not affect p53 protein function, the ratio of total p53 and acetylated p53 would still remain the same regardless of whether p53 is tagged or not. However, the issue is more complicated given that different levels of p53 occupy promoters of different target genes (due to differences in promoter affinity). Therefore it is formerly likely that at higher total p53 levels, there will also be more Halo-tagged p53 bound to “lower affinity” promoters (that would not have been occupied at low total p53 levels), therefore it then becomes possible for the alteration in resident time of p53 to alter due to alterations in dissociation constants at different promoter strengths. An example of promoter affinity induced gene expression alteration could be the result of selective binding of RNA-Pol II (Morachis et al 2010 Genes & Development). While I am aware of the Supplementary Figure with H1299 cells, I recommend using K382R mutant in the p53 shRNA MCF7 background, and inducing the expression of the K382R mutant to different levels (such as different hours of Dox induction, 2h versus 4h for example). Any difference in p53 behavior would then suggest that acetylation of K382 is not the sole determinant of p53 bound fraction/residence time. Alternatively to overcome the issue of differences of promoter affinity, western blot showing total levels of both transiently affected Halo-tagged wt and K382Q p53 as well as endogenous p53 level would suffice as performed in Fig. 4c.

3. Inconclusive evidence to support causality of p53 binding kinetics to its transcriptional activity:

This point directly addresses the title of the paper chosen by the authors. Assuming point 1 and 2 are addressed, and that indeed, the acetylation of p53 is the sole determinant of p53 bound fraction and residence time, it does not necessarily mean that residence time is causative to altered p53 transcriptional activity. The altered transcriptional activity may also result from factors not discussed by the author such as tetramerization rates of p53, or alterations in co-factor binding resulting from alternative post-translational modifications not studied by the authors. In the absence of the complete scenario, it would only be possible to draw a correlation but not causation between p53 residence time and the actual transcription of downstream target genes. Proving causation in this scenario may be difficult, as any mutation block/mimic studies would not only disrupt the p53 binding kinetics the authors describe but simultaneously also alter any potential co-factor interactions or tetramerization properties of p53. However, should point 1 and 2 be addressed, the authors can undoubtedly still claim causation of p53 acetylation leading to changes in p53 bound fraction and residence time, whilst the link between p53 residence time to

target gene transcription may at best remain as a correlation. In this case the authors should tone down their conclusions and title.

Reviewer #2 (Remarks to the Author):

The molecular mechanisms that regulate the binding kinetics of transcription factors within living cells remains poorly understood. Furthermore, whether and how the binding kinetics modulates transcription remains largely unclear. By using the state-of-art live-cell single molecule imaging, Jacchetti et al. reported that the binding kinetics of p53, a critical tumor suppressor, can be modulated by genotoxic stress. They further demonstrated that the binding kinetics of p53 can be regulated by post-transnational modification (PTM), acetylation. Crucially, their data suggested that the residence time rather than abundance of p53 correlates with the induction of target genes. These novel single-molecule results represent a significant step in our further understanding of how PTM controls the transactivating activity of transcription factor by modulating its binding kinetics. The conclusions are quantitatively supported by their data. There are some concerns needed to be addressed before acceptance.

1. Jacchetti et al. extracted kinetic fractions of p53 by decomposing the histogram of single-molecule displacement. Since their SMT was the 2-D projection of the 3-D motion of individual molecules, there is bias toward slowly moving particles. Although the reported fractions have been corrected via Monte-Carlo simulation, the correction should be verified by experimental methods. One way is to count only the first displacement of each track.
2. As for analyzing the average free time between two binding events, the theoretic basis should be clearly specified. It is also unclear what assumption has been made.
3. Page 3, line 73 and 74, the statement is confusion. It contradicted with Figure 1a where the abundance of HaloTag-p53 is elevated by genotoxic stress. Such elevation may not be due to the increased expression of the fusion gene.
4. In the method section, there are lots of jargon needed to be fixed.
5. Figure 3 is not clear.

Reviewer #3 (Remarks to the Author):

Review of “C-terminal acetylation controls the p53 single molecule binding kinetics in living cells to modulate its transcriptional activity”

Overall assessment

This paper by Jacchetti, Mazza and co-workers presents a nice SPT study of HaloTag-p53

building on work by Mazza and Morisaki in the McNally lab. They identify, through a number of perturbation experiments, an interesting link between p53 acetylation and residence time. Overall, the work is a nice contribution that I think should eventually be published in Nature Communications. However, I have a number of issues which should be addressed convincingly first, which are listed below. Most importantly, I have serious concerns about how they track fast-diffusing molecules and the authors must do another experiment at a faster frame rate (at least 100 Hz) to avoid catastrophic tracking errors. Also importantly, the authors overstate the link between their nice work on p53 residence time/acetylation and transcriptional output. They have not demonstrated a causal link and their manuscript and title should be change to reflect what they have shown instead of what they would like to show. Once these issues have been properly addresses, and if the main conclusions stand, I believe this will make a nice contribution to Nature Communications.

Tracking diffusing molecules

The authors track both bound and diffusing molecules. They use this to calculate diffusion constants, search times, fraction bound etc. According to the text, they image at a frame rate of 10 frames/sec (10 Hz) using 5 ms stroboscopic laser excitation pulses. The stroboscopic illumination is important because it overcomes the problem that fast-diffusing molecules otherwise “spread” their photons over many pixels and will be missed by most localization algorithms. However, since there is a gap of 100 ms between frames, fast-diffusing molecules can move quite far in that period even if they are precisely localized. In Mazza 2012, the p53 diffusion constant was reported as 3.4 $\mu\text{m}^2/\text{s}$. In 100 ms, the mean displacement would be $\sqrt{4 \cdot 3.4 \cdot 0.1} = 1.2 \mu\text{m}$. However, a large fraction of molecules would move more than 2-3 μm and a rare few would move the diameter of the nucleus. Based on the example images, such as in Fig. 1c, there is thus no way that they can perform the tracking without accumulating many tracking errors over the course of a movie. This may not affect the estimation of residence times, since here there is little movement estimation of the free fraction and its diffusion constant it is stroboscopic. E.g. in Fig 1c it looks like they have between 5-20 molecules in focus per cell. The bound fraction is small, so the majority is undergoing rapid diffusion. Thus, it is impossible to confidently connect localizations in consecutive trajectories without major errors. This unfortunately invalidates the modeling of displacements in Fig 1d. Thus, an essential revision – and I cannot recommend publication in any journal without this – is for the authors to repeat the tracking using a faster frame rate (e.g. 100 or 200 Hz) with their stroboscopic illumination and then repeat the fitting of displacements. In this case, the density of Halo-TMR molecules must be lower so as to reduce tracking errors.

I note that the reported p53 diffusion constant is 1.86 here, in stark contrast with the value of 3.4 previously reported in Mazza 2012. 3.4 is much more realistic – 1.86 is very slow and much slower than what one would expect from a TF. The authors fail to address this discrepancy anywhere in the text, but such a difference may be explained by the problems with the tracking in this manuscript.

Modeling displacements

In Mazza, NAR, 2012, a very nice and rigorous method for modeling displacements was introduced and very clearly explained. A substantially different and seemingly less rigorous method is used in this paper. The authors should explain why. For example, the authors ignore the effect of localization errors on the displacement histograms even though it is very easy to correct for (as in Mazza 2012) – why? In Mazza 2012, the probability of a molecule switching between states during the observation time was explicitly accounted for, but not in this paper – why? I could not find anywhere how many consecutive localizations they consider for calculating displacement (e.g. frame n to $n+1$, or n to $n+2$, or all or what?). The probability that a molecule remains in focus after 100 ms is quite small, which is a problem. Moreover, details were somewhat lacking on how the observation volume correction was done. As a reader, $D=3.4$ was reported in 2012 and $D=1.8$ is reported now, which is of some concern and the method used now is seemingly less rigorous. Please explain.

Average residence time

Line 185-188: 3.1 s +/- residence time of p53 is in excellent agreement with previous studies. Ref 20 reports 3.3 sec and 2.5 sec, Mueller (Ref 21) also found around 2-3 sec and Morisaki found 3.5 and 5.5 sec. Most people that do double-exponential fitting report the fast and the slow component. But here the authors report the “average” of the slow and fast component. In fact, the slow component is quite a bit slower (6-12 sec or so). The authors should more clearly distinguish between the fast vs. slow (or specific vs. non-specific), since the interpretation of these two (e.g. sliding on DNA vs. functional binding at REs) is quite different.

Moreover, and this was another point I was a bit confused by, how is average res time calculated? I assume the authors take all of their observations and plot the 1-CDF and then do 2-exp fitting. The average res time is then $= F_{\text{fast}} * \text{ResTime}_{\text{fast}} + F_{\text{slow}} * \text{ResTime}_{\text{slow}}$ from all of their single-molecule observations.

Now, I have one concern with this. Suppose a hypothetical TF, XTF, has a very short fast component: 10 ms and a very long slow component 100 s. So the fast component is 10,000x faster. Suppose the TF spends half its bound time in non-specific fast interaction and the other half in slow, specific interactions. Now, for every slow trajectory you should have 10,000 fast ones. So, the average res time is $= F_{\text{fast}} * \text{ResTime}_{\text{fast}} + F_{\text{slow}} * \text{ResTime}_{\text{slow}} = 99.99\% * 10 \text{ ms} + 0.01\% * 100\text{s} = 20 \text{ ms}$. However, this number is not an accurate descriptor of the fact that XTF spends half its time in long (100 sec) and half its time in very short (10 ms) binding. Now obviously, this XTF example is a bit extreme, but ideally the method should be robust regardless of the numbers. It is also possible I misunderstood their calculation – can the authors clarify this?

Search times

Line 203-209: the use of the p53 binding mutant is an important control. However, I was

confused by their calculation of search times (Line 210-215; but most details in supplement). Their approach seems to be based on Ref 28. Now I am not saying that their approach is wrong, I am just saying that I don't fully understand it from their methods or Ref 28's methods and would like them to explain their calculation much more clearly in their revision and in their response to this. Here is another, much simpler calculation. Let F be the fraction bound, then $F = k_{ON}/(k_{ON}+k_{OFF})$. Let τ be the search time. Then $\tau = 1/k_{ON} = (F*k_{OFF})/(1-F_{bound})$. We want τ for specific sites. From Fig S2, I estimate F for wt to be 35% after IR and 28% for mSB-mutant-p53. This would suggest that the fraction bound to specific sites is 7%. The residence time at specific sites appears to be around 12 sec. So plugging in the numbers: $\tau = (1-0.07) / (0.07*(1/12 \text{ s})) = 160$ seconds or about 2.5 minutes. The search time is listed as being 5 min in Fig 2d.

Can the authors explain why they got a different number than me? If they believe their method is correct, can they explain it in more detail and explain why the above simple calculation is incorrect?

Kymograph analysis of residence times

The senior author, Mazza, published a really nice paper on p53 in *Nucleic Acids Research* 2012. In this paper, "objective thresholding" is used – the residence time as analyzed as a function of N , where N is the minimum number of frames the molecule was observed. The residence time is then calculation as a function of N and N is determined by the point of convergence. This seemed solid and Mazza carefully cross-validated those results using FRAP and FCS. However, the kymograph analysis seems similar in some respect – in the supplement it is written that a minimum number of frames is also used, however, no reason for these choices are given. They may have been carefully chosen, but no reason is given. The authors should explain in detail in the supplement why they made the choices that they did and also they should explain why they do not do "objective thresholding" anymore in the supplement and ideally show the same data analyzed with both methods.

Degradation of p53

As a TF, p53 is one of the more important which is probably why the authors chose to study it. However, its regulation is quite complicated – it is activated in dynamic pulses that can vary substantially from cell-to-cell (see Lahav lab papers). Moreover, p53 is constantly made and degraded. Thus, one concern I had would be if HaloTag-p53 is labelled with TMR and then degraded. When the proteasome gets to the last amino acids with the TMR ligand what then happens? Does the TMR-peptide hang around the cell? And how can the authors distinguish between real p53-TMR and partially degraded p53-TMR? It is maybe too technical for the main text, but it is very important for the interpretation of the experiments, so the authors should address this in the supplement at least.

It would also be nice to explain in more detail the dynamics of HaloTag-p53 in response to 10 Gy IR – does this result in similar dynamics to UV irradiation (see Purvis & Lahav, *Cell*, 2013,

Figure 2D). I know the authors show Westerns in Fig 1a which is nice, but it would be nice if they could quantify it on a plot to make it more clear for the reader to assess the extent to which HaloTag-p53 dynamics follow endogenous wt-p53 dynamics. It is tough for the reader to quantify-by-eye the WB.

Also, why did the authors use irradiation rather than nutlin treatment for p53 activation (just for my curiosity: what the authors did was fine)?

Acetylation and residence time: effect on transcription

In Figure 3, evidence is provided that C-terminal acetylation on two lysines increases the residence time in a Wortmannin-dependent fashion. However, the Wortmannin both changes acetylated p53 level and total p53 level. The authors should include in Fig. 3 a Western quantification showing the fraction p53 that is acetylated before and after Wortmannin treatment.

The authors attempt to link p53 residence time to transcription output. This is an important goal for the field, but extremely difficult technically, and the authors do not have the evidence. The smFISH of the p53-target CDKN1 is correlative. Linking gene output to p53 residence time is extremely complicated. First of all, different genes (e.g. early vs. late) read out p53 dynamics differently (see work by Lahav lab). Second, 10 Gy IR activates a large number of pathways and it is not totally clear to which extent CDKN1 output is related to p53. Third, nascent CDKN1 mRNA peaks at 2 h, but since so many things change dynamically, it is not clear to which extent this is caused by acetylation and residence time. Thus, the authors need to substantially tone down the language linking p53 dynamics and acetylation to transcription. They have identified an interesting correlation, but not a causal link. For example, in line 360-362:

“To the best of our knowledge, this is the first time that modulation of transient TF residence times in living cells and transcriptional output have been connected for natural eukaryotic binding sites, although similar correlations have been shown to hold for other TFs at artificial arrays of binding sites.”

What the authors say here is not true. They have not proven that increasing p53 residence time at the CDKN1 promoter increases transcriptional output. Specific deletion of p53 binding elements from the CDKN1 locus would be a necessary experiment to directly link p53 activity to CDKN1 transcriptional output. Thus, this sentence has to be removed. If anything, the paper published in 2014 in Nature Communications by Morisaki et al. makes a strong link between residence time and transcription.

Additionally, the authors' finding that CDKN1 transcription is modulated by the number of active gene loci rather than the number of nascent transcripts seems to contradict their model in which p53 residence time is the primary driver of CDKN1 expression. Such a model fails to explain how two nearly identical allelic copies can be differentially regulated based solely on the residence time of p53. Can the authors clarify in the text?

To sum up: the authors have found an interesting seemingly causal connection between CTD

acetylation and residence time, though it is unclear which is causing the other one. The authors have not established any causal link between acetylation/residence time and transcriptional output and they must change the paper title and paper text to make clear what they have and have not shown.

We are grateful to the referees for the insightful comments on our manuscript. We have now performed additional experiments to address their concerns and we have modified the manuscript accordingly. Both reviewer #1 and #3 have raised the observation that our model connecting p53 binding kinetics, p53 CTD acetylation and p53 mediated transcriptional activation is based on correlative data, and that the discussion of this model should be toned down. While providing additional data to support that p53 acetylation controls transcriptional activation of p53 target genes, we agree with the referees and we have restated the title and the discussion of our paper accordingly. The updated text in the revised version of the manuscript and of the supplementary information is highlighted in **bold**.

Reviewers' comments:

Reviewer #1 (Remarks to the Author):

A longstanding question in the p53 field revolves around how specific post-translational modifications govern the transcriptional output at the molecular level, which enables p53 to govern the downstream cellular response accordingly. In this paper, Jacchetti et al describes a novel molecular mechanism that links acetylation of p53, a known post-translational modification to p53 following irradiation, to the increased resident time of p53 on DNA and target gene promoters. Should this observation hold true, it does provide new insights into the roles of acetylation in regulating the promoter binding by transcription factors. However, as it stands, the results included are insufficient to enable the authors to draw the overarching conclusions, as several key experiments and controls are missing.

We are grateful to the reviewer for highlighting important weaknesses of the previous version of our manuscript. We have tried to address the points raised by the reviewer, by performing additional experiments and by rewriting parts of our discussion. A point-by-point reply is attached below.

1. Inconclusive evidence to support that p53 levels alone do not alter bound fraction of p53:

The authors conclude that p53 level alone does not cause a change in bound fraction of p53 (Line 228-230), which would be reasonable based on the effects of doxycycline induced only at 2 hours (Fig 3a-c). However this conclusion is difficult to draw for 4 hours of Doxycycline addition as the total p53 has changed drastically. Furthermore, at the 24 hours Doxycycline addition time point, where a large change in p53 bound fraction is observed, there is no Western blot to accompany the graph to conclude whether it is a change in p53 level or acetylation that is causing the difference, or indeed other factors/reasons, such as activation of proteins that in turn modulate p53 behavior.

Using Wortmannin as an indirect drug to inhibit K382 acetylation, the authors claim that the reduced acetylation at K382 (Line 236-239, Fig 3d-f) led to reduced p53 bound fraction. However in this case, although the Western blot shows a substantial reduction in K382-acetylation, the total level of p53 is equally reduced, therefore the alteration in p53 behavior may also arise as a result of p53 level, and not just due to acetylation. The same argument applies to Fig.4, where although the authors elegantly use Set8 knock down as well as acetylation mimic/blocking (KQ/KR) mutants to show the effect of acetylation on p53 bound fraction, the total level of p53 was not quantified. In order to rule out the potential effect of p53 levels, a Western blot is necessary to show that the total p53 level (regardless of its acetylation status/mutation) is unaltered.

We agree that in our original manuscript the evidence for the independence of p53 binding kinetics on p53 levels was incomplete. We have now performed additional experiments to show that an increase of p53

levels is not sufficient to induce the stabilization of p53 binding. First we add a western blot for the treatment of MCF-7 with 24hrs Doxycycline (Figure 3 a-c). Both the p53 levels and p53 acetylation further increase at 24hrs Doxy.

We have also performed experiments stratifying our SMT data on p53 based on the nuclear fluorescence intensity of HaloTag-p53 (Supplementary Figure 5), and we now show that only a small difference (< 6%) in bound fraction is measured for cells expressing HaloTag-p53-wt (in particular bright cells display a slightly lower bound fraction than dim ones). Importantly (see below) the bound fraction of mutants inhibiting p53 acetylation is independent on the protein levels.

Finally, we have added western blots to the experiments in Fig 4, to show that silencing of Set8 does not result in a modulation of p53 levels and that transiently transfected wt or mutant HaloTag-p53 all share the same expression levels, with the sole exception of the 6Q mutant.

Taken together this data rule out that the increase in p53 binding observed upon activation by DNA damage is due to the increase in p53 levels.

2. Inconclusive evidence to support that promoter affinity does not alter resident time of p53:

This is an important point given that the authors have overexpressed Halo-tagged p53 in the background of endogenous p53, which poses a further complication on p53 level analysis. In all Western blots, the authors have only shown the Halo-tagged p53, but not the endogenous p53 in cells. It is possible that both the level and acetylation status of endogenous p53 levels have changed too. One can assume that should the Halo-tag not affect p53 protein function, the ratio of total p53 and acetylated p53 would still remain the same regardless of whether p53 is tagged or not.

We now have added blots for the total and acetylation levels of untagged p53 and their quantification, to show that the ratio between acetylation and total p53 is similar for the endogenous population and for the HaloTagged one, in cell populations undergoing irradiation and wortmannin treatment. Cells treated with Doxycycline for 4 and 24hrs show an enrichment in total levels and acetylation for the tagged population only, possibly due to the low levels of expression of the endogenous protein, that might not allow reliable quantification of the acetylated population.

However, the issue is more complicated given that different levels of p53 occupy promoters of different target genes (due to differences in promoter affinity). Therefore it is formerly likely that at higher total p53 levels, there will also be more Halo-tagged p53 bound to "lower affinity" promoters (that would not have been occupied at low total p53 levels), therefore it then becomes possible for the alteration in resident time of p53 to alter due to alterations in dissociation constants at different promoter strengths.

An example of promoter affinity induced gene expression alteration could be the result of selective binding of RNA-Pol II (Morachis et al 2010 Genes & Development). While I am aware of the Supplementary Figure with H1299 cells, I recommend using K382R mutant in the p53 shRNA MCF7 background, and inducing the expression of the K382R mutant to different levels (such as different hours of Dox induction, 2h versus 4h for example). Any difference in p53 behavior would then suggest that acetylation of K382 is not the sole determinant of p53 bound fraction/residence time. Alternatively to overcome the issue of differences of promoter affinity, western blot showing total levels of both transiently affected Halo-tagged wt and K382Q p53 as well as endogenous p53 level would suffice as performed in Fig. 4c.

The point raised here by the reviewer is very important, as it is possible that low-affinity promoters get occupied when increasing the p53 levels. However, the expectation in this case would be that the average residence time of p53 would decrease with the increase of p53 levels (as lower affinity corresponds to a

larger k_{off} – which is the inverse of the residence time). This effect would be in the opposite direction to what we observe upon activation of p53 by irradiation: the p53 levels increase and the p53 binding increases as well.

To rule out that the saturation of high affinity binding sites could strongly bias our analysis, we performed experiments similar to what proposed by the referee to further exclude that the modulation of p53 binding is due to a modulation in p53 levels. First, we now show that transiently transfected H1299 (the one used for the experiments on mutants, to avoid the formation of hetero-tetramers) with either WT- or K382mut-HaloTag-p53 all share the same expression levels of the transfected protein (Fig 4a). Second, we generated MCF-7 knocked out for endogenous p53 by CRISPR/Cas9 and transiently transfected them with HaloTag p53-wt or HaloTag p53-K382R (Supplementary Figure 5). Also in this case, the p53 binding was modulated upon DNA damage only for p53-wt but not for the acetylation-blocked mutant. We stratified cells depending on the expression levels of the transfected protein and we found no difference in the p53-K382R mutant binding when looking at cells expressing the transgenes at lower or at higher levels.

3. Inconclusive evidence to support causality of p53 binding kinetics to its transcriptional activity:

This point directly addresses the title of the paper chosen by the authors. Assuming point 1 and 2 are addressed, and that indeed, the acetylation of p53 is the sole determinant of p53 bound fraction and residence time, it does not necessarily mean that residence time is causative to altered p53 transcriptional activity. The altered transcriptional activity may also result from factors not discussed by the author such as tetramerization rates of p53, or alterations in co-factor binding resulting from alternative post-translational modifications not studied by the authors. In the absence of the complete scenario, it would only be possible to draw a correlation but not causation between p53 residence time and the actual transcription of downstream target genes. Proving causation in this scenario may be difficult, as any mutation block/mimic studies would not only disrupt the p53 binding kinetics the authors describe but simultaneously also alter any potential co-factor interactions or tetramerization properties of p53. However, should point 1 and 2 be addressed, the authors can undoubtedly still claim causation of p53 acetylation leading to changes in p53 bound fraction and residence time, whilst the link between p53 residence time to target gene transcription may at best remain as a correlation. In this case the authors should tone down their conclusions and title.

This is another point that is very important and discussed also by Ref #3.

We agree that our evidence for connecting p53 binding kinetics and transcriptional activity is of a correlative nature and we have now rewritten the title and the discussion of the manuscript to better reflect our data. At the same time we have added some smFISH data to provide more convincing evidence that p53 acetylation is important for transcriptional activation, by looking in p53-knocked out MCF-7 cells for the ability of transiently transfected p53-wt and p53-K382R to activate CDKN1a transcription. We are currently setting up a method to probe p53 binding and p53 tetramerization simultaneously, that we hope will provide more insights on the interplay of these two phenomena in controlling p53-mediated transcription, which will be the object of a future study.

Reviewer #2 (Remarks to the Author):

The molecular mechanisms that regulate the binding kinetics of transcription factors within living cells remains poorly understood. Furthermore, whether and how the binding kinetics modulates transcription remains largely unclear. By using the state-of-art live-cell single molecule imaging, Jacchetti et al. reported that the binding kinetics of p53, a critical tumor suppressor, can be modulated by genotoxic stress. They

further demonstrated that the binding kinetics of p53 can be regulated by post-translational modification (PTM), acetylation. Crucially, their data suggested that the residence time rather than abundance of p53 correlates with the induction of target genes. These novel single-molecule results represent a significant step in our further understanding of how PTM controls the transactivating activity of transcription factor by modulating its binding kinetics. The conclusions are quantitatively supported by their data. There are some concerns needed to be addressed before acceptance.

We are grateful to the reviewer for his/her positive comments on the manuscript. We have tried to address his questions as detailed below.

1. Jacchetti et al. extracted kinetic fractions of p53 by decomposing the histogram of single-molecule displacement. Since their SMT was the 2-D projection of the 3-D motion of individual molecules, there is bias toward slowly moving particles. Although the reported fractions have been corrected via Monte-Carlo simulation, the correction should be verified by experimental methods. One way is to count only the first displacement of each track.

We agree with the referee that the analysis could bias towards slowly moving particles and that in the original version of the manuscript we have not provided evidence that the MC simulation approach correctly takes care of this potential problem. However, we believe that the method proposed by the referee would still show some bias as the first displacement of each track might still be affected by the underestimation of fast-moving molecules, and would potentially bias our analysis towards freely diffusing molecules, as they re-enter the observation volume more frequently than bound ones. We have approached the problem with a different experiment, though. We have repeated one measurement by acquiring data at a much faster frame rate (100Hz). If the MC correction is ineffective we would have expected to measure different bound fractions when acquiring at different frame rates. The bound fractions are instead in excellent agreement, underlying the validity of the correction approach. We also include in the supplementary materials the validation of our correction by performing MC simulations.

2. As for analyzing the average free time between two binding events, the theoretic basis should be clearly specified. It is also unclear what assumption has been made.

We have now better described the calculation of free times and search times in the supplementary materials. The derivation of the free time between binding events as a function of bound fraction and average residence time assumes that the system is at equilibrium and the reactions follow first order kinetics. However we argue that the same relationship would hold if the binding process can be considered ergodic or in other words, if the bound fraction can be equally calculated by measuring which fraction of molecules are bound at any time, or by counting the fraction of time that each molecule stays bound for. Notably, stimulated by the referee question and by a point raised by referee 3 we have now improved our calculation of search times, and provide evidence of the correctness of our approach by simulations in the response to referee 3.

3. Page 3, line 73 and 74, the statement is confusing. It contradicted with Figure 1a where the abundance of HaloTag-p53 is elevated by genotoxic stress. Such elevation may not be due to the increased expression of the fusion gene.

We apologize for the confusing statement. We now clarified that both endogenous and tagged p53 levels are elevated by genotoxic stress, due to the disruption of the feedback loop between p53 and its negative regulator MDM2.

4. In the method section, there are lots of jargon needed to be fixed.

We tried to better describe our experimental procedures in the method section of the revised manuscript.

5. Figure 3 is not clear.

We now provide a higher quality image for figure 3. We apologize for the inconvenience.

Reviewer #3 (Remarks to the Author):

Review of “C-terminal acetylation controls the p53 single molecule binding kinetics in living cells to modulate its transcriptional activity”

Overall assessment

This paper by Jacchetti, Mazza and co-workers presents a nice SPT study of HaloTag-p53 building on work by Mazza and Morisaki in the McNally lab. They identify, through a number of perturbation experiments, an interesting link between p53 acetylation and residence time. Overall, the work is a nice contribution that I think should eventually be published in Nature Communications. However, I have a number of issues which should be addressed convincingly first, which are listed below. Most importantly, I have serious concerns about how they track fast-diffusing molecules and the authors must do another experiment at a faster frame rate (at least 100 Hz) to avoid catastrophic tracking errors.

Also importantly, the authors overstate the link between their nice work on p53 residence time/acetylation and transcriptional output. They have not demonstrated a causal link and their manuscript and title should be change to reflect what they have shown instead of what they would like to show.

Once these issues have been properly addresses, and if the main conclusions stand, I believe this will make a nice contribution to Nature Communications.

We are grateful to the referee for the detailed and careful review of our manuscript. We have tried to address his concerns as detailed below.

Tracking diffusing molecules

The authors track both bound and diffusing molecules. They use this to calculate diffusion constants, search times, fraction bound etc. According to the text, they image at a frame rate of 10 frames/sec (10 Hz) using 5 ms stroboscopic laser excitation pulses. The stroboscopic illumination is important because it overcomes the problem that fast-diffusing molecules otherwise “spread” their photons over many pixels and will be missed by most localization algorithms. However, since there is a gap of 100 ms between frames, fast-diffusing molecules can move quite far in that period even if they are precisely localized. In Mazza 2012, the p53 diffusion constant was reported as $3.4 \mu\text{m}^2/\text{s}$. In 100 ms, the mean displacement would be $\sqrt{4 \cdot 3.4 \cdot 0.1} = 1.2 \mu\text{m}$. However, a large fraction of molecules would move more than 2-3 μm and a rare few would move the diameter of the nucleus. Based on the example images, such as in Fig. 1c, there is thus no way that they can perform the tracking without accumulating many tracking errors over the course of a movie. This may not affect the estimation of residence times, since here there is little movement estimation of the free fraction and its diffusion constant it is stroboscopic. E.g. in Fig 1c it looks like they have between 5-20 molecules in focus per cell. The bound fraction is small, so the majority is undergoing rapid diffusion. Thus, it is impossible to confidently connect localizations in consecutive trajectories without major errors. This unfortunately invalidates the modeling of displacements in Fig 1d. Thus, an essential

revision – and I cannot recommend publication in any journal without this – is for the authors to repeat the tracking using a faster frame rate (e.g. 100 or 200 Hz) with their stroboscopic illumination and then repeat the fitting of displacements. In this case, the density of Halo-TMR molecules must be lower so as to reduce tracking errors. I note that the reported p53 diffusion constant is 1.86 here, in stark contrast with the value of 3.4 previously reported in Mazza 2012. 3.4 is much more realistic – 1.86 is very slow and much slower than what one would expect from a TF. The authors fail to address this discrepancy anywhere in the text, but such a difference may be explained by the problems with the tracking in this manuscript.

We agree with the referee that the risk of mistracking/losing particles increases when using long frame rates. For a diffusion coefficient D the molecule has a probability of diffusing more than a certain displacement r equal in a time t equal to $e^{-(r^2/4Dt)}$. So for $t = 100\text{ms}$ and $D = 1.86\mu\text{m}^2/\text{s}$ (the diffusion coefficient we measured in this work, more on the diffusion coefficient will be discussed in the following) the probability of moving more than $2\mu\text{m}$ is less than 1%. If we take a value of $D = 3.4\mu\text{m}^2/\text{s}$ the probability of moving more than $2\mu\text{m}$ rises to about 5%. A 5% error would result in a reshuffling of a 5% of the distribution of jumps corresponding to free molecules. However, even if mistracked, these molecules would still be counted as free ones, therefore this error would not significantly affect the estimation of the bound fraction. Further, on average, the number of tracked molecules per cell per frame was of about 5, therefore the average distance between molecules is typically larger than $2\mu\text{m}$.

Nevertheless, we agree that it is desirable to validate our method by performing the experiments at different frame rates, as suggested by the reviewer. In our 2012 Nucleic Acid Research we have shown that the measured bound fraction for p53 was equal within the experimental error at the chosen frame rates (varied between 50 and 10 Hz, Supplementary Figure 2c), while the diffusion coefficient of the free populations was varying (from 1.7 to 3.4 $\mu\text{m}^2/\text{s}$), and we attributed this difference to the nature of diffusion in the nuclear environment, which is not free, but hindered by the high molecular crowding. In the NAR paper we tested this hypothesis by modelling the mean squared displacement curve of the free p53 molecules which was found to follow an anomalous behavior with exponent α equal to 0.82.

Following the comments of the reviewer, we decided to perform single molecule tracking at a faster frame rate (100Hz) and with shorter illumination time (2 ms) in a representative condition (namely upon stabilization of p53 binding by genotoxic stress). At this fast frame rate, the contribution of the different populations (bound, slow diffusing, fast diffusing) to the distribution of displacements overlap significantly, and we therefore used mathematical modelling of the full distribution of displacements (as described in our NAR paper) to estimate the bound fraction and the diffusion coefficients of the mobile populations. The data has been now added to the supplementary material (Supplementary Figure 2c), showing that the estimated bound fraction obtained at 100Hz is comparable to the one obtained at 10Hz (25% +/- 3% at 100Hz, compared to 21% +/- 2 at 10Hz). The estimated diffusion coefficient of the diffusing populations is instead significantly larger at 100Hz than at 10Hz (2.89 vs 1.79 $\mu\text{m}^2/\text{s}$ for the fast diffusing population). Interestingly the slowly diffusing population shows a similar trend, suggesting that these differences might not be due to mistracking (which should mostly affect the fast diffusing population). In accordance to our NAR paper, we rather interpret these diffusion coefficients as apparent diffusion coefficients that change over time due to anomalous diffusion, described by the law:

$\text{MSD} = 4 * D(t) * t$ with $D(t) = A * t^{(\alpha-1)}$ → the apparent diffusion coefficient at different frame rates is then calculated as $D(t = 100\text{ms}) = D(t = 10\text{ms}) * 10^{(\alpha-1)}$ and if we use $\alpha = 0.82$ and $D(t = 10\text{ms}) = 2.89$ we estimate $D(t = 100\text{ms}) = 1.85$, in excellent agreement with the diffusion coefficient measured at this frame rate.

Modeling displacements

In Mazza, NAR, 2012, a very nice and rigorous method for modeling displacements was introduced and very clearly explained. A substantially different and seemingly less rigorous method is used in this paper. The authors should explain why.

The referee is right in saying that the method used for quantifying the bound fraction – based on the fitting of the distribution of displacements between one frame and the next - is simpler and makes more assumptions than the complete model used in our 2012 NAR paper, which based on the modelling of the full spatio-temporal distribution of displacements (that is the distribution of displacements for Δt , $2\Delta t$, etc) and could be used to estimate both the bound fraction and the residence time. The reason for these simplifications, now described in the supplementary materials, is essentially one: The full spatio-temporal distribution of displacements is much more sensitive to tracking errors. To better explain the problem imagine to have a tracking algorithm that has an error rate of 1%, for example given by the possibility that 1 time every 100 the fluctuations in the photon emission of the single molecule would render the image dimmer than our detection threshold. In this case, when analyzing displacements at $\Delta t = 1$ frame only 1% of the displacements will be affected by the mistracking. However, when analyzing displacements at $\Delta t = 100$ frames, basically all the tracked segments will be affected by the tracking error. The problem is particularly relevant for the estimation of residence times. In our case we measured an average residence time of 6 seconds (that is 60 frames): if we were to apply the full model to fit the spatio-temporal distribution of displacements – given a tracking error of 1% - we would underestimate the residence time for a large fraction of bound molecules. In the 2012 NAR paper we solved this issue by manually checking the tracks of bound molecules one by one. Here we wanted to avoid manual intervention given the number of different conditions tested, and to avoid biases in the analysis. Therefore we split the problem in two: we analyzed the bound fraction with a simple model (with its approximations, see below) and the residence time with an algorithm that does not require tracking but is based segment detection in kymographs.

Regarding the approximations of the method used to calculate the bound fraction these deals mostly with the assumption that a small fraction of molecules exchange from the bound and the free state in Δt and that the diffusion coefficients allow for separation between the different populations at $\Delta t = 100$ ms. These points are now detailed in the supplementary information of the manuscript and below.

For example, the authors ignore the effect of localization errors on the displacement histograms even though it is very easy to correct for (as in Mazza 2012) – why?

True, the effect of the localization error was dismissed in the original manuscript as it only affects the estimates of the diffusion coefficient by a small amount. The D originally reported in our manuscript were affected by the localization precision σ by $D_{\text{measured}} = D_{\text{true}} + \sigma^2/\Delta t$. We now have updated the manuscript to include a measurement of the localization precision (Supplementary Figure 2e) and the measured diffusion coefficients have been updated accordingly.

In Mazza 2012, the probability of a molecule switching between states during the observation time was explicitly accounted for, but not in this paper – why?

True. The switching between bound and free state has not been accounted in this simplified model. For our estimates of residence times t_b longer than 3s, the probability of a molecule of switching from a bound state to a free state is roughly: $\exp(-\Delta t/t_b) < 4\%$. Further these rapid unbinding events contribute to the distribution of displacements with one displacement each, while an event lasting 30 frames contributes with 29 displacements. Therefore the underestimation of the bound fraction less 2×10^{-3} , much lower than the statistical error associated to the bound fraction itself and we think that it can be reasonably dismissed. These arguments are now presented in the supplementary materials.

I could not find anywhere how many consecutive localizations they consider for calculating displacement (e.g. frame n to $n+1$, or n to $n+2$, or all or what?).

We now better clarify in the text that we consider the displacements from frame n to $n + 1$.

The probability that a molecule remains in focus after 100 ms is quite small, which is a problem. Moreover, details were somewhat lacking on how the observation volume correction was done. As a reader, $D=3.4$ was reported in 2012 and $D=1.8$ is reported now, which is of some concern and the method used now is seemingly less rigorous. Please explain.

The probability for a molecule at the focal plane to remain within detection area in 100ms is roughly $\text{erf}(\Delta z/\sqrt{4*D*\Delta T})$, so for the fast moving molecules, $D = 1.8\mu\text{m}^2/\text{s}$ and $\Delta z = 0.75\mu\text{m}$, this probability is about 80%. In practice the situation is a little bit more complex as the molecule could start from any z within the detection slice. In our 2012 NAR paper, we have dealt with this issue by modelling the edges of the detection slice as absorbing boundaries (molecule lost when touches the edge of the detection slice). This correction does not account however for the fact that during the blind time between two acquisitions a molecule could escape and re-enter the detection slice – so without further correction the correction factor is overestimated. To provide correct estimates for the correction factor we therefore performed Monte Carlo simulations to quantify the fraction of molecules with a given diffusion coefficient that are lost because going out of focus in 100ms, and we corrected the fractions measured for this factor. The methods for the MC simulation are now explained in more detail in the supplementary materials section. Reassuringly, since we measured a similar bound fraction at faster frame rates, analyzed with the more rigorous model, we now provide experimental evidence that the MC correction is properly correcting for the loss of molecules that exit the detection slice.

Average residence time

Line 185-188: 3.1 s +/- residence time of p53 is in excellent agreement with previous studies. Ref 20 reports 3.3 sec and 2.5 sec, Mueller (Ref 21) also found around 2- 3 sec and Morisaki found 3.5 and 5.5 sec. Most people that do double-exponential fitting report the fast and the slow component. But here the authors report the “average” of the slow and fast component. In fact, the slow component is quite a bit slower (6-12 sec or so). The authors should more clearly distinguish between the fast vs. slow (or specific vs. non-specific), since the interpretation of these two (e.g. sliding on DNA vs. functional binding at REs) is quite different.

Moreover, and this was another point I was a bit confused by, how is average res time calculated? I assume the authors take all of their observations and plot the 1-CDF and then do 2-exp fitting. The average res time is then $= F\text{-fast} * \text{ResTime}\text{-fast} + F\text{-slow} * \text{ResTime}\text{-slow}$ from all of their single-molecule observations.

Now, I have one concern with this. Suppose a hypothetical TF, XTF, has a very short fast component: 10 ms and a very long slow component 100 s. So the fast component is 10,000x faster. Suppose the TF spends half its bound time in non-specific fast interaction and the other half in slow, specific interactions. Now, for every slow trajectory you should have 10,000 fast ones. So, the average res time is $= F\text{-fast} * \text{ResTime}\text{-fast} + F\text{-slow} * \text{ResTime}\text{-slow} = 99.99\% * 10 \text{ ms} + 0.01\% * 100\text{s} = 20 \text{ ms}$. However, this number is not an accurate descriptor of the fact that XTF spends half its time in long (100 sec) and half its time in very short (10 ms) binding. Now obviously, this XTF example is a bit extreme, but ideally the method should be robust regardless of the numbers. It is also possible I misunderstood their calculation – can the authors clarify this?

The referee is right regarding how the average residence time is calculated. The average thereby reflects not only the characteristic times of the two populations but also how frequent it is to find molecules in one of the two particular states. In the example of the referee, the fast events are 10000x more frequent than the slow ones and therefore they weigh more on the average. In our opinion this is what is measured in ensemble average techniques when multiple components cannot be discriminated (i.e. Mueller et al 2008, Hinow et al 2006, Mazza et al 2012): frequent events will be weighted more than infrequent ones in providing the average residence time.

Search times

Line 203-209: the use of the p53 binding mutant is an important control. However, I was confused by their calculation of search times (Line 210-215; but most details in supplement). Their approach seems to be based on Ref 28. Now I am not saying that their approach is wrong, I am just saying that I don't fully understand it from their methods or Ref 28's methods and would like them to explain their calculation much more clearly in their revision and in their response to this. Here is another, much simpler calculation. Let F be the fraction bound, then $F = k_{ON}/(k_{ON}+k_{OFF})$. Let Tau be the search time. Then $\text{Tau} = 1/k_{ON} = (F \cdot k_{OFF})/(1-F_{\text{bound}})$. We want Tau for specific sites. From Fig S2, I estimate F for wt to be 35% after IR and 28% for mSB-mutant-p53. This would suggest that the fraction bound to specific sites is 7%. The residence time at specific sites appears to be around 12 sec. So plugging in the numbers: $\text{Tau} = (1-0.07) / (0.07 \cdot (1/12 \text{ s})) = 160 \text{ seconds}$ or about 2.5 minutes.

The search time is listed as being 5 min in Fig 2d.

Can the authors explain why they got a different number than me? If they believe their method is correct, can they explain it in more detail and explain why the above simple calculation is incorrect?

We are grateful to the referee for the point raised, as it has allowed us to identify an inaccuracy in the original derivation of the calculation of search times by Chen et al., 2014, that we inadvertently carried over in our manuscript. We have corrected the calculation as described below, importantly while the resulting search times change significantly (they drop to about 100s), the main result of figure 2d (last panel) hold: the p53 search time is not modulated upon the induction of DNA damage.

We prefer to use our calculation over the one proposed by the referee (that seems reasonable) for two main reasons. First, the referee calculation is based on the results obtained on the cell line H1299 (no endogenous p53), which is not the system that has been used for the calculation of search times before and after the induction of DNA damage (MCF-7), which is considered a more physiological system for p53 dynamics (H1299 cells might have adapted to not have p53). While the modulation of p53 binding is observed both in MCF-7 and H1299, the absolute numbers are somewhat different, and this might affect the estimates. Second, the referee calculation depends on comparing the result of p53-wt to that of our mutant in the DNA binding domain. While the main effect of this mutant is to abolish specific p53 binding (Morisaki et al., 2014) the mutations on the p53 surface interacting with DNA might also – to some extent – affect non-specific interactions.

For these reasons we favor the method described by Chen et al., as it allows to obtain an estimate of the search time from one single experiment. However as noted above we now identified an inaccuracy in the original method:

The method calculates the search time by first estimating how frequent is on average the association to specific binding sites compared to all binding events, which is given by:

$$P_{B,s} = \frac{k_{on,s}}{k_{on,s} + k_{on,ns}}$$

Where $k_{on,s}$ and $k_{on,ns}$ are the pseudo-association rates to long-lived and short-lived binding sites. For example if $P_{B,s} = 0.1$, on average there will be one binding event to long-lived sites for every 10 binding events, or in other words one molecule will bind for 10 times before finding a specific site. Therefore we can define the average number of trials to find a specific site as $N_{trials} = \frac{1}{P_{B,s}}$.

Once N_{trials} is obtained, the total time for finding a specific site will be given by N_{trials} multiplied by the time it takes on average to make one of these trials. This time is composed by the time in which the

molecule is diffusing in the nucleoplasm τ_{3D} plus the time in which the molecule is bound nonspecifically τ_{ns} . Since the last of the trials correspond to finding the specific site, we subtract $1 \tau_{ns}$ to calculate the total search time $\tau_{search} = N_{Trials}\tau_{3D} + (N_{Trials} - 1)\tau_{ns}$.

In Chen et. al, the authors try to calculate $P_{B,s}$ from the fraction of binding events measured at specific sites from the biexponential fit of the 1-CDF distribution of residence times F_s , arguing that:

$$F_s = \frac{k_s \tau_s}{k_s \tau_s + k_{ns}\tau_{ns}}$$

We now realize that this is not the right definition for F_s : the quantity described above is a measure of how likely it is to find at one particular time one molecule bound at a specific site rather than at a non-specific site. Instead, the F_s extracted by the fitting of the 1-CDF do not depend on the duration of binding events and are instead directly a measure of how frequent are association events at specific sites. In other word a right mathematical expression for F_s is:

$$F_s = P_{B,s} = \frac{k_{on,s}}{k_{on,s} + k_{on,ns}}$$

To support our claim, we performed MCs simulations for molecules diffusing in a box of 10 μm in edge, and reversibly binding to two species of binding sites, with binding parameters $k_{on,s}, k_{off,s}$ for the specific binding sites and $k_{on,ns}, k_{off,ns}$ for the non-specific binding sites. We repeated the simulations by varying

$k_{on,s}$ over 4 order of magnitude, and then identified the binding events and populated a 1-CDF distribution of binding sites. We then fitted these distributions with a biexponential to extract estimates for $k_{off,s}, k_{off,ns}$, and F_s .

The results of the simulations, shown on the left display that:

1. We can obtain correct estimates for the residence times (inverse of the fitted k_{off}) for a broad range of $P_{B,s}$, ranging from 2.5% to 95% (left inset panel).

2. The fitted F_s is identical to the imposed

$P_{B,s} = \frac{k_{on,s}}{k_{on,s} + k_{on,ns}}$ in the simulations (right inset panel).

Simulations of the distribution of residence times for two component binding. We run multiple simulations by fixing the dissociation rates of bound molecules to $k_{off,s}$ and $k_{off,ns}$. The association rate to non-specific binding sites was set to $k_{on,ns}$, while the association rate to specific binding sites was varied between 0.01 and 100. From the simulations we extracted the duration of binding events which were then fitted by a biexponential decay to extract estimates for the k_{off} (left inset) and for the fraction of long-lived binding events F_s (right inset).

We now have updated the methods describing the calculation of the search times in the supplementary material, and the corresponding results in the main text (figure 2d - last panel).

Kymograph analysis of residence times

The senior author, Mazza, published a really nice paper on p53 in *Nucleic Acids Research* 2012. In this paper, “objective thresholding” is used – the residence time as analyzed as a function of N, where N is the minimum number of frames the molecule was observed. The residence time is then calculation as a function of N and N is determined by the point of convergence. This seemed solid and Mazza carefully cross-validated those results using FRAP and FCS. However, the kymograph analysis seems similar in some respect – in the supplement it is written that a minimum number of frames is also used, however, no reason for these choices are given. They may have been carefully chosen, but no reason is given. The authors should explain in detail in the supplement why they made the choices that they did and also they should explain why they do not do “objective thresholding” anymore in the supplement and ideally show the same data analyzed with both methods.

We now describe more in detail the reasoning behind the thresholds chosen in our kymograph approach. The conditions chosen to identify the bound molecules by the kymograph method satisfy the “objective thresholding method”: to have a probability that a free molecule diffusing with a $D=0.2 \text{ um}^2/\text{s}$ will be erroneously counted as bound less than 1%. We also show that the kymograph approach provides a distribution of residence times that is overlapping with the one measured by tracking molecules and applying our original “objective thresholding” method.

Degradation of p53

As a TF, p53 is one of the more important which is probably why the authors chose to study it. However, its regulation is quite complicated – it is activated in dynamic pulses that can vary substantially from cell-to-cell (see Lahav lab papers). Moreover, p53 is constantly made and degraded. Thus, one concern I had would be if HaloTag-p53 is labelled with TMR and then degraded. When the proteasome gets to the last amino acids with the TMR ligand what then happens? Does the TMR-peptide hang around the cell? And how can the authors distinguish between real p53-TMR and partially degraded p53-TMR? It is maybe too technical for the main text, but it is very important for the interpretation of the experiments, so the authors should address this in the supplement at least.

We are grateful to the referee for this insightful comment. The question raised is difficult to answer, but it is very relevant given that inactive p53 is rapidly degraded through the MDM2 mediated negative feedback loop. We have tried to minimize the potential problem of tracking partially degraded p53 by performing the labelling as close as possible to the observation by single molecule microscopy (that is half an hour before).

We now add additional experiments to show that partially degraded p53 or unligated TMR do not contribute to the tracked molecules. First, we performed western blots using an antibody recognizing HaloTag in MCF7/6/HP53 cells (Supplementary figure S2). In the presence of degradation products that still carry a functional tag, we would expect to see a smear or additional bands at lower molecular weight. These degradation products constitute only a small fraction of the HaloTag p53 band, indicating that degradation products do not likely affect our analysis. We also repeated the experiment by looking directly at the fluorescence of the ligand on the blot. In these conditions we could appreciate only one additional band, that was also visible in the control lane where parental MCF-7 cells (that do not carry HaloTag p53) were incubated with the fluorescent ligand. To rule out that such unligated/non specifically bound ligand could be tracked, we performed single molecule tracking after adding the fluorescent ligand to parental cells. In these conditions we could not track single molecules in the nucleus of MCF-7 cells with the current settings of the microscope, possibly because the ligand by itself moves too fast (motion blur) or is kept in cytosolic compartments (for example in acidic compartments such as lysosomes).

It would also be nice to explain in more detail the dynamics of HaloTag-p53 in response to 10 Gy IR – does this result in similar dynamics to UV irradiation (see Purvis & Lahav, *Cell*, 2013, Figure 2D). I know the

authors show Westerns in Fig 1a which is nice, but it would be nice if they could quantify it on a plot to make it more clear for the reader to assess the extent to which HaloTag-p53 dynamics follow endogenous wt-p53 dynamics. It is tough for the reader to quantify-by-eye the WB.

We now have added the quantification of WB in figure 1a. The dynamics of HaloTag-p53 reflect those of endogenous p53 in the same cell line, and they both seem slightly delayed compared to the dynamics of p53 in the parental cell line, possibly due to the moderate over-expression of HaloTag-p53 in our cells.

Also, why did the authors use irradiation rather than nutlin treatment for p53 activation (just for my curiosity: what the authors did was fine)?

The effects of Nutlin-3 on transcriptional activation on a short time-scale is elusive, with some studies showing that no transactivation of canonical genes can be detected with conventional RT-PCR after short term treatment with nutlin (Allen et al. 2014). We have preliminary data that shows that in our cells upon nutlin treatment p53 binding and p53 transcriptional activation are upregulated. However, given these controversies, we would like to explore the effects of nutlin more in detail in future projects.

Acetylation and residence time: effect on transcription

In Figure 3, evidence is provided that C-terminal acetylation on two lysines increases the residence time in a Wortmannin-dependent fashion. However, the Wortmannin both changes acetylated p53 level and total p53 level. The authors should include in Fig. 3 a Western quantification showing the fraction p53 that is acetylated before and after Wortmannin treatment.

The quantification of the ratio acetylation/total p53 for the wortmannin treatment has now been added in Supplementary Figure 4b. Clearly the fraction of acetylated p53 drops significantly following treatment with wortmannin.

The authors attempt to link p53 residence time to transcription output. This is an important goal for the field, but extremely difficult technically, and the authors do not have the evidence.

We agree that our attempt to connect p53 binding kinetics and transcriptional output is only based on correlative evidence. While we have performed some additional experiments to further try to support our model (see below), we have substantially modified the title and the discussion of our paper to highlight the limits of this analysis and the future work that needs to be performed in our opinion to provide a definitive proof of our model.

The smFISH of the p53-target CDKN1 is correlative. Linking gene output to p53 residence time is extremely complicated. First of all, different genes (e.g. early vs. late) read out p53 dynamics differently (see work by Lahav lab).

We agree that different genes read out p53 dynamics differently, and we discuss the limitation of our "genome-wide" measurement of binding in the updated discussion of our paper. However, we would also like to note that recent data from a previous post-doc from the Lahav lab seem to support that the differences in gene expression between early and late p53 regulated genes seem to be controlled at post-transcriptional level, by tuning the mRNA decay rate of transcribed genes (Porter, Fisher, and Batchelor 2016).

Second, 10 Gy IR activates a large number of pathways and it is not totally clear to which extent CDKN1 output is related to p53.

We now have added a FISH experiment on MCF7 cells knocked out for p53 (Figure S6c), to show that 2hrs post IR the CDKN1a transcription is abolished in absence of p53.

Third, nascent CDKN1 mRNA peaks at 2 h, but since some many things change dynamically, it is not clear to which extent this is caused by acetylated and residence time.

To provide additional evidence for the role of acetylated p53 in causing activation of CDKN1a transcription by rescuing MCF7/p53KO cells with transiently transfected p53-wt or an unacetylatable mutant (p53-K382R). While cells transfected with p53wt resulted in reactivation of CDKN1a transcription upon irradiation, those transfected with p53-K382R did not.

Thus, the authors need to substantially tone down the language linking p53 dynamics and acetylation to transcription. They have identified an interesting correlation, but not a causal link. For example, in line 360-362:

“To the best of our knowledge, this is the first time that modulation of transient TF residence times in living cells and transcriptional output have been connected for natural eukaryotic binding sites, although similar correlations have been shown to hold for other TFs at artificial arrays of binding sites.”

What the authors say here is not true. They have not proven that increasing p53 residence time at the CDKN1 promoter increases transcriptional output. Specific deletion of p53 binding elements from the CDKN1 locus would be a necessary experiment to directly link p53 activity to CDKN1 transcriptional output. Thus, this sentence has to be removed. If anything, the paper published in 2014 in Nature Communications by Morisaki et al. makes a strong link between residence time and transcription.

We have removed the sentence and rewritten the discussion to tone down the connection between the increase in residence times and the activation of transcription.

Additionally, the authors' finding that CDKN1 transcription is modulated by the number of active gene loci rather than the number of nascent transcripts seems to contradict their model in which p53 residence time is the primary driver of CDKN1 expression. Such a model fails to explain how two nearly identical allelic copies can be differentially regulated based solely on the residence time of p53. Can the authors clarify in the text?

While we understand the concern of the referee, we do not agree with the statement that CDKN1a transcription is modulated by the number of active loci would contradict our model. smFISH provides just a snapshot of the active transcription occurring at the time of fixation. It is therefore possible that at the moment of fixation only one copy of the two allelic genes is “stably” bound at the promoter by p53. In agreement with this model Sencal et al. (Senecal et al., Cell Rep., 2014) have recently shown that the residence time of artificial transcription factors to the Fos enhancer modulate the duration of transcriptional bursts, rather than the number of elongating polymerases on the gene. In a snapshot picture as the one provided by FISH this correspond to having longer TF residence times that translate into a larger fraction of active transcription sites rather than in a higher number of nascent transcripts per transcription site. We have now clarified this point in the results and in the discussion sections of the manuscript.

To sum up: the authors have found an interesting seemingly causal connection between CTD acetylation and residence time, though it is unclear which is causing the other one. The authors have not established any causal link between acetylation/residence time and transcriptional output and they must change the paper title and paper text to make clear what they have and have not shown.

We have modified the title and the text of our paper according to the suggestions of the referee. We point out that our mutational analysis argue for p53 acetylation controlling the p53 binding kinetics.

Reviewers' Comments:

Reviewer #1 (Remarks to the Author):

I wish to congratulate the authors for putting together such a comprehensive revision, which satisfactory addressed my main concerns, including the confounding effect of total p53 level on their measurements of DNA residence time. The additional western blots and controls presented in this revision are also important and convincing. I have only 2 very minor comments regarding the choice of words in the introduction (see below). Other than that I don't have other major concerns and I support publication.

1) Lines 92-95: "Immunoprecipitation experiments have also failed in showing a modulation in total p53 promoter occupancy following the activation by DNA damage as ChIP show only small changes in p53 occupancy at target genes following p53 activation, attributed to the change in p53 expression levels. Ref[16] ". This is confusing. The paper cited shows that changes in p53 ChIP signal are proportional to total changes in protein levels (range from 2-6 fold) and that they see no evidence for p53 PTMs modulating p53 DNA binding. The authors should reformulate their writing to fit better with the conclusions of the cited paper and be more specific.

2) Line 111: The authors write: "To this end we prove that (i)...(ii)...(iii)..." The word prove is too strong here. I suggest changing it to "show"

Reviewer #2 (Remarks to the Author):

The authors have addressed my concerns.

One minor point:

In supplementary methods, the references should be listed.

Reviewer #3 (Remarks to the Author):

Second review of "C-terminal acetylation controls the p53 single molecule binding kinetics in living cells to modulate its transcriptional activity"

Overall assessment

The revised manuscript by Jacchetti, Mazza and co-workers is much improved and it is clear that the authors have made a valiant effort and included much-needed new experiments and analysis. The revised manuscript presents a lot of interesting data and important new results, and is very careful in not over-interpreting the results. I still have a few quibbles, which I provide for the information of the authors below.

The revised calculation of the search times is also important and correct the problems with Chen et al. Cell 2014. I also note that the correction has resulted in a big change in the search time estimate and I find the revised search times much more plausible. I suspect that localization errors will dominate the estimates of the fast component of the 2-exp fit to 1-CDF of binding times, which will affect the search time, but I agree that the logic of the author's calculation is correct and thank the authors for providing a clear explanation of their revised calculations.

I still disagree with the way the authors are doing their tracking. I looked at their supplementary movies and with the long gap between frames (10 Hz) and the relatively high density of molecules, it is simply impossible in many cases to link molecules unambiguously between frames (What the authors calculation of fraction below $2\mu\text{m}$ fails to account for is out-of-focus molecules moving into focus very close to another molecule in focus). Therefore, strictly speaking, the authors are not really doing "single-particle tracking". Nevertheless, I acknowledge that even if 2 different molecules are linked into the same trajectory frequently, this will almost certainly be 2 different free molecules and thus, the bound fraction would not be strongly affected. I was also not convinced by the anomalous MSD calculation because the logic is circular: incorrect tracking leads to underestimates of displacements. Underestimates of displacements compounding over time leads to artefactual subdiffusive alpha. Incorrect subdiffusive alpha is then used to rationalize incorrect D at 10 Hz. At 10 Hz, a major contribution to apparent subdiffusive alpha also comes from confinement inside nucleus.

Moreover, the authors claim that the probability of a molecule moving out-of-focus in 100 ms is only 20%: but this is obviously wrong because the diffusion constant is underestimated and because the molecule could be anywhere in Z. If you take all of this into account and assume an observation slice of 700-800 nm, I estimate around 60-70% will be lost, which is huge. The authors now set their observation slice to 1.5 μm , which I do not find convincing (it is too large).

Dear Editor and referees,

We have modified the manuscript according to the last comments of the referees, as detailed below.

Reviewer #1 (Remarks to the Author):

I wish to congratulate the authors for putting together such a comprehensive revision, which satisfactorily addressed my main concerns, including the confounding effect of total p53 level on their measurements of DNA residence time. The additional western blots and controls presented in this revision are also important and convincing. I have only 2 very minor comments regarding the choice of words in the introduction (see below). Other than that I don't have other major concerns and I support publication.

1) Lines 92-95: "Immunoprecipitation experiments have also failed in showing a modulation in total p53 promoter occupancy following the activation by DNA damage as ChIP show only small changes in p53 occupancy at target genes following p53 activation, attributed to the change in p53 expression levels. Ref[16] ". This is confusing. The paper cited shows that changes in p53 ChIP signal are proportional to total changes in protein levels (range from 2-6 fold) and that they see no evidence for p53 PTMs modulating p53 DNA binding. The authors should reformulate their writing to fit better with the conclusions of the cited paper and be more specific.

2) Line 111: The authors write: "To this end we prove that (i)...(ii)...(iii)..." The word prove is too strong here. I suggest changing it to "show".

We are grateful to the reviewer for his/her comments. We have modified the text of our manuscript according to the referee suggestions: in particular (1) Line 91-94. The sentence has been modified to: "Also, according to immunoprecipitation experiments, p53 binding to REs seem mainly tuned by modulating p53 expression levels, rather than by post-translational modifications, as in response to DNA damage the fold-change in p53 occupancies closely match the changes in the TF expression." (2) Line 110 We have changed "prove" to "show".

Reviewer #2 (Remarks to the Author):

The authors have addressed my concerns.

One minor point:

In supplementary methods, the references should be listed.

We are grateful to the reviewer for his/her comments. We have added the reference list to the supplementary materials.

Reviewer #3 (Remarks to the Author):

Overall assessment

The revised manuscript by Jacchetti, Mazza and co-workers is much improved and it is clear that the authors have made a valiant effort and included much-needed new experiments and analysis. The revised manuscript presents a lot of interesting data and important new results, and is very careful in not over-interpreting the results. I still have a few quibbles, which I provide for the information of the authors below.

The revised calculation of the search times is also important and correct the problems with Chen et al. Cell 2014. I also note that the correction has resulted in a big change in the search time estimate and I find the revised search times much more plausible. I suspect that localization errors will dominate the estimates of the fast component of the 2-exp fit to 1-CDF of binding times, which will affect the search time, but I agree that the logic of the author's calculation is correct and thank the authors for providing a clear explanation of their revised calculations.

I still disagree with the way the authors are doing their tracking. I looked at their supplementary movies and with the long gap between frames (10 Hz) and the relatively high density of molecules, it is simply impossible in many cases to link molecules unambiguously between frames (What the authors calculation of fraction below 2um fails to account for is out-of-focus molecules moving into focus very close to another molecule in focus). Therefore, strictly speaking, the authors are not really doing "single-particle tracking". Nevertheless, I acknowledge that even if 2 different molecules are linked into the same trajectory frequently, this will almost certainly be 2 different free molecules and thus, the bound fraction would not be strongly affected.

We are grateful for the comment of the referee, we now state more clearly that at lower frame rates there might be mislinking between different tracks, but this does not seem to affect the estimation of the bound fraction (Lines 181-186).

I was also not convinced by the anomalous MSD calculation because the logic is circular: incorrect tracking leads to underestimates of displacements. Underestimates of displacements compounding over time leads to artefactual subdiffusive alpha. Incorrect subdiffusive alpha is then used to rationalize incorrect D at 10 Hz. At 10 Hz, a major contribution to apparent subdiffusive alpha also comes from confinement inside nucleus.

While we agree with the comment of the referee, we wanted to point out that the measurement of the subdiffusive anomalous exponent performed in our NAR paper was carried out at a faster frame rate (50 to 25Hz), thereby mitigating some of the possible artifacts in the calculation. Nevertheless, we now point out in the Supplementary Information that the inconsistency in the diffusion coefficients measured at 10 and at 100Hz could also be due to mistracking at the lower frame rate. We tried to estimate what would be the impact on the measured by numerical simulations, as detailed below.

Moreover, the authors claim that the probability of a molecule moving out-of-focus in 100 ms is only 20%: but this is obviously wrong because the diffusion constant is underestimated and because the molecule could be anywhere in Z. If you take all of this into account and assume an observation slice of 700-800 nm, I estimate around 60-70% will be lost, which is huge. The authors now set their observation slice to 1.5 um, which I do not find convincing (it is too large).

We are grateful for the comment. We first point out that numerical simulations for $D = 1.82 \text{ um}^2/\text{s}$ and for $\Delta Z = 1.5 \text{ um}$ that explicitly account for the molecule being "anywhere in Z" show that the uncorrected bound fraction would be overestimated by around 20%, in agreement with the back-of-the-envelope calculation provided in the first response to the referee. Therefore, the effect of the molecule being anywhere in Z is a relatively small one.

As we discussed above, we agree with the referee that it is possible that our estimation of the diffusion coefficient for the free population is under-estimated due to mistracking. Similarly, the measured observation slice thickness by imaging beads might be overestimated (due to the larger signal to noise ratio observed for beads compared than the one achievable for single molecules in the nucleus of living cells).

We therefore tried to quantify how these uncertainties would affect the estimation of our bound fractions. To this scope we first roughly quantified the thickness of the observation slice by collecting 3D stacks of individual TMR molecules in fixed cells. We point out that this measurement is approximate, since (i) the labelling density and the presence of out-of-focus signal might strongly affect this measurement. (ii) We have observed that the photon-count associated to individual TMR molecules in fixed cells is lower than in live cells (iii) Molecules might photobleach during the z-stack, resulting in an underestimation of the observation slice thickness. With these caveats, we measured a Δz ranging between 0.85 and 1.35 μm .

We next simulated what would be the apparent bound fraction at 10Hz, using a diffusion coefficient for the free molecules equal to $2.9 \mu\text{m}^2/\text{s}$, an imposed bound fraction of 50% and a $\Delta z = 1 \mu\text{m}$. The uncorrected fitted bound fraction, resulted equal to 69% (that is an overestimation of 37%). Applying the correction approach used in the paper – that is with $D = 1.86$ and $\Delta z = 1.5$, resulted in a corrected bound fraction equal to 60% (20% overestimation).

Throughout our manuscript we measured experimental bound fractions ranging from 10% to 25%. Therefore, the net effect of a possible underestimation of D and a possible overestimation of Δz ranges from 2% to 5% on the measured bound fractions. We now added these additional comments to the supplementary materials of our paper.